# Hdac6 regulates Tip60-p400 function in stem cells

**Poshen B Chen**[1]**, Jui-Hung Hung**[2†]**, Taylor L Hickman**[3]**, Andrew H Coles**[1]**, James F Carey**[1]**, Zhiping Weng**[2,4]**, Feixia Chu**[3]**, Thomas G Fazzio**[1,5]*****

[1]Program in Gene Function and Expression, University of Massachusetts Medical School, Worcester, United States; [2]Program in Bioinformatics and Integrative Biology, University of Massachusetts Medical School, Worcester, United States; [3]Department of Molecular, Cellular and Biomedical Sciences, University of New Hampshire, Durham, United States; [4]Department of Biochemistry and Molecular Pharmacology, University of Massachusetts Medical School, Worcester, United States; [5]Program in Molecular Medicine, University of Massachusetts Medical School, Worcester, United States

*****For correspondence: thomas. fazzio@umassmed.edu

[†]**Present address:** Department of Biological Science and Technology and Institute of Bioinformatics and System Biology, National Chiao Tung University, Taiwan

**Competing interests:** The authors declare that no competing interests exist.

**Reviewing editor**: Kevin Struhl, Harvard Medical School, United States

**Abstract** In embryonic stem cells (ESCs), the Tip60 histone acetyltransferase activates genes required for proliferation and silences genes that promote differentiation. Here we show that the class II histone deacetylase Hdac6 co-purifies with Tip60-p400 complex from ESCs. Hdac6 is necessary for regulation of most Tip60-p400 target genes, particularly those repressed by the complex. Unlike differentiated cells, where Hdac6 is mainly cytoplasmic, Hdac6 is largely nuclear in ESCs, neural stem cells (NSCs), and some cancer cell lines, and interacts with Tip60-p400 in each. Hdac6 localizes to promoters bound by Tip60-p400 in ESCs, binding downstream of transcription start sites. Surprisingly, Hdac6 does not appear to deacetylate histones, but rather is required for Tip60-p400 binding to many of its target genes. Finally, we find that, like canonical subunits of Tip60-p400, Hdac6 is necessary for robust ESC differentiation. These data suggest that Hdac6 plays a major role in the modulation of Tip60-p400 function in stem cells.

## Introduction

ESC self-renewal and differentiation are controlled by multiple pathways: exogenous factors that act through well-defined signaling pathways that are also employed in adult cells, and a network of nuclear factors that regulate the ESC transcriptome (*Hanna et al., 2010*). Regulators of gene expression can be further sub-divided into (i) sequence-specific transcription factors, including ESC-specific master regulators, (ii) non-coding RNAs that act both in cis and in trans to regulate specific subsets of genes, and (iii) chromatin regulatory complexes, most of which are expressed in multiple cell and tissue types, and often act very broadly in the genome to covalently modify histones, remodel nucleosomes, or modify higher-order chromatin folding (*Hanna et al., 2010*; *Young, 2011*). A number of chromatin regulators have been identified from RNA-interference screens or traditional knockout studies that are important for various features of ESC identity. However, for most chromatin regulatory complexes, several key questions remain, including how they find their genomic targets, how their catalytic activities lead to alteration of gene expression, and how the activities of these factors are altered to facilitate differentiation.

In mammals, several chromatin remodeling complexes are modular, with distinct forms expressed in different cell or tissue types, or sometimes within the same cells (*Wang et al., 1996*; *Ho et al., 2009*; *Ramírez and Hagman, 2009*; *Fazzio and Panning, 2010*; *Hanna et al., 2010*). For example, the mammalian SWI/SNF-family complex BAF (Brg1/Brahma Associated Factor) consists of several related

**eLife digest** Embryonic stem cells are cells that are able to transform into many other types of cells, such as blood cells and skin cells, as well as being able to divide in order to produce more stem cells. Mature cells lack this ability, which is called pluripotency, which is why there is so much interest in using embryonic stem cells to replace or regenerate human cells that have been lost or damaged through injury or illness.

The various processes that result in self-renewal (the production of new stem cells) or differentiation (the production of other types of cells) are controlled by a wide variety of pathways, including some that only apply to the regulation of gene expression in stem cells. A number of these processes are known to involve chromatin – the densely packed structure formed by DNA and proteins called histones.

Now Chen et al. study the means by which chromatin controls the stem cell fates by examining how a large enzyme called Tip60-p400 that interacts with histones – one of the main components of chromatin – in both mature cells and embryonic stem cells. Tip60-p400 is known to switch on genes that cause stem cells to undergo self-renewal, and to switch off the genes that allow stem cells to transform into other cell types, but the molecular mechanisms responsible for these effects have not yet been identified.

Chen et al. studied the activity of Tip60-p400 in mouse embryonic stem cells, and found that another enzyme, Hdac6, had to be present for Tip60-p400 to regulate the genes in the stem cells. Hdac6 is mostly found in the cytoplasm of cells that have differentiated into other cell types, and in the nucleus of stem cells, which is where the DNA resides. In cells from mice that lack Hdac6, Chen et al. also found that stem cells fail to replicate *or* differentiate properly in culture, underscoring the importance of this particular enzyme, and filling in another piece of the puzzle of stem cell biology.

complexes with many shared subunits, plus a few subunits that are specific to each particular cell type. In particular, when neural progenitors differentiate into neurons in mouse, two BAF subunits are replaced with two paralogous subunits that shift BAF from a factor promoting self-renewal to one that promotes differentiation (*Lessard et al., 2007*; *Yoo et al., 2009*; *Hanna et al., 2010*; *Young, 2011*). Another unique combination of subunits, different from those observed in differentiated cells, comprises BAF complex from ESCs (esBAF) (*Ho et al., 2009*). Similarly, multiple forms of PRC1 (Polycomb Repressive Complex 1) have been purified from human and mouse cells that each contain the Ring1a/b ubiquitin ligase, but have different arrays of accessory proteins that confer distinct target specificity and activities (*Gao et al., 2012*; *Tavares et al., 2012*).

Tip60-p400 has been purified from cancer cell lines as a 17 subunit chromatin remodeling complex with two chromatin remodeling activities: the Tip60 (also known as Kat5) subunit acetylates the N-terminal tails of histones H2A, H4, and a number of transcription factors, while the p400 subunit mediates exchange of H2A–H2B dimers for H2AZ–H2B dimers within nucleosomes (*Doyon et al., 2004*; *Cai et al., 2005*; *Squatrito et al., 2006*). In somatic cells, Tip60-p400 serves mainly as a transcriptional co-activator that functions with numerous sequence-specific transcription factors to activate gene expression (*Brady et al., 1999*; *Baek et al., 2002*; *Frank et al., 2003*; *Legube et al., 2004*). In contrast, while Tip60-p400 promotes expression of some genes required for cellular proliferation and cell cycle regulation in ESCs, its most prominent function is to silence genes that are active during differentiation (*Fazzio et al., 2008a*, *2008b*). RNAi-mediated knockdown (KD) of several Tip60-p400 subunits in ESCs individually induces a phenotype in which differentiation and ESC markers are expressed simultaneously, proliferation is reduced, the cell cycle is altered, and cells exhibit diminished self-renewal and pluripotency (*Fazzio et al., 2008a*). Consistent with these phenotypes, mice homozygous for a *Tip60* deletion allele die at the pre-implantation stage (*Hu et al., 2009*). It remains unknown why Tip60-p400 functions mainly as a repressor of differentiation gene expression in ESCs rather than an activator of expressed genes, as it does in most cell types examined.

Similarly, treatment of ESCs with Trichostatin A (TSA), a drug that broadly inhibits class I and II HDACs and results in elevated acetylation of most lysines targeted by HATs, promotes morphological changes similar to those observed upon KD of Tip60-p400 subunits (*McCool et al., 2007*; *Karantzal et al., 2008*). Therefore, maintenance of proper levels of histone acetylation appears to be essential

to perpetuate the pluripotent state, as neither significant increases nor decreases in histone acetylation appear to be compatible with ESC self-renewal. However, TSA also inhibits several HDAC family members known to target acetylated lysines on non-histone proteins, leaving open the possibility that these targets play an equal or greater role in maintenance of ESC self-renewal. Furthermore, deletion or KD of several individual HDACs in ESCs produces phenotypes that differ substantially from those of Tip60-p400 subunits (*McBurney et al., 2003*; *Dovey et al., 2010*), suggesting that different HDACs perform different functions in ESC self-renewal.

In this study, we interrogate the composition of Tip60-p400 complex in mouse ESCs in order to identify unique interacting proteins that might account for its altered functional repertoire in this cell type. We find that the class II histone deacetylase (HDAC), Hdac6, is a stable interaction partner with Tip60-p400 in ESCs, but not mouse embryo fibroblasts (MEFs). Subsequent analyses revealed that Hdac6 also interacts with Tip60-p400 in adult neural stem cells from the brain and some cancer cell lines, but is sequestered away from Tip60-p400 in the cytoplasm of most differentiated cell types, as previously reported (*Verdel et al., 2000*; *Hubbert et al., 2002*; *Kawaguchi et al., 2003*; *Valenzuela-Fernández et al., 2008*). We show that Hdac6 is necessary for regulation of most Tip60-target genes in ESCs, particularly differentiation genes repressed by Tip60-p400 in ESCs. Surprisingly, while its deacetylase domains are required for silencing of differentiation genes, Hdac6 does not regulate gene expression by deacetylating histones near the promoters of Tip60-p400 targets. Instead, the catalytic domains of Hdac6 are required for its interaction with Tip60-p400. Furthermore, we find that Hdac6 is necessary for normal Tip60-p400 enrichment at its gene targets, just downstream of their transcription start sites (TSSs), suggesting that Hdac6 helps recruit Tip60-p400 complex to many target gene promoters. Finally, we show that Hdac6 is necessary for several major functions of Tip60-p400 in ESCs, as both *Tip60-* and *Hdac6*-deficient ESCs have defects in formation of single colonies, reduced proliferation rates, and defects in differentiation. However, unlike *Tip60* KD, *Hdac6* KD does not prevent ESC self-renewal. Thus, Hdac6 is a component of a novel, stem cell-specific, form of Tip60-p400 complex that is necessary for gene regulation and normal differentiation in ESCs.

## Results

### A class II HDAC, Hdac6, co-purifies with Tip60-p400 complex in ESCs and NSCs

Tip60-p400 complex has roles in both activation and repression of transcription in most cell types where it has been examined. However, in ESCs, Tip60-p400 is required for repression of many more genes than it activates, raising the possibility that a unique form of Tip60-p400 complex that might be expressed in ESCs that shifts the balance of its activity toward a more repressive role. To test this possibility, we targeted a 36 amino acid 6-histidine-3-FLAG (H3F) tag to the C-terminus of one copy of the endogenous *Tip60* gene in murine ESCs (*Figure 1—figure supplement 1*), performed double affinity purifications from tagged or untagged cells (*Figure 1A*), and identified proteins that co-purified with Tip60 using LC-MS/MS (*Table 1*). By this approach, we identified 16 of 17 known subunits (*Ikura et al., 2000*; *Cai et al., 2003*, *2005*; *Doyon et al., 2004*; *Altaf et al., 2009*) of Tip60-p400, suggesting that expression of the tagged form of Tip60 from its endogenous locus allowed for normal complex formation. Furthermore, we observed a number of novel Tip60-p400-interacting proteins, including chromatin regulatory proteins and transcription factors. To test for cell type specificity of Tip60-p400 complexes we generated a knock-in mouse harboring *Tip60-H3F*, isolated embryonic fibroblasts and repeated the purification. We observed several bands within Tip60 purifications from ESCs that were not observed in purifications from *Tip60-H3F* MEFs or untagged cells (*Figure 1A*), consistent with the possibility that ESCs express a distinct form of Tip60-p400 complex.

We were intrigued by the finding that Hdac6 co-purified with Tip60-p400 in ESCs (*Table 1*). Hdac6 is a class II histone deacetylase (HDAC) that is expressed in many different cell types but is usually localized to the cytoplasm (*Verdel et al., 2000*; *Hubbert et al., 2002*; *Kawaguchi et al., 2003*; *Valenzuela-Fernández et al., 2008*), as are its well-established substrates: α-tubulin (*Hubbert et al., 2002*), Hsp90 (*Kovacs et al., 2005*), and cortactin (*Zhang et al., 2007*). Moreover, despite its homology to proteins that deacetylate histone tails, Hdac6 has not been found to target histones in vivo (*Haggarty et al., 2003*). To confirm that Hdac6 is a bona fide Tip60-p400-interacting protein, we performed reciprocal co-immunoprecipitation experiments in ESCs, observing the Tip60-Hdac6 interaction no matter which protein was immunoprecipitated (*Figure 1B*, *Figure 1—figure supplement 2*).

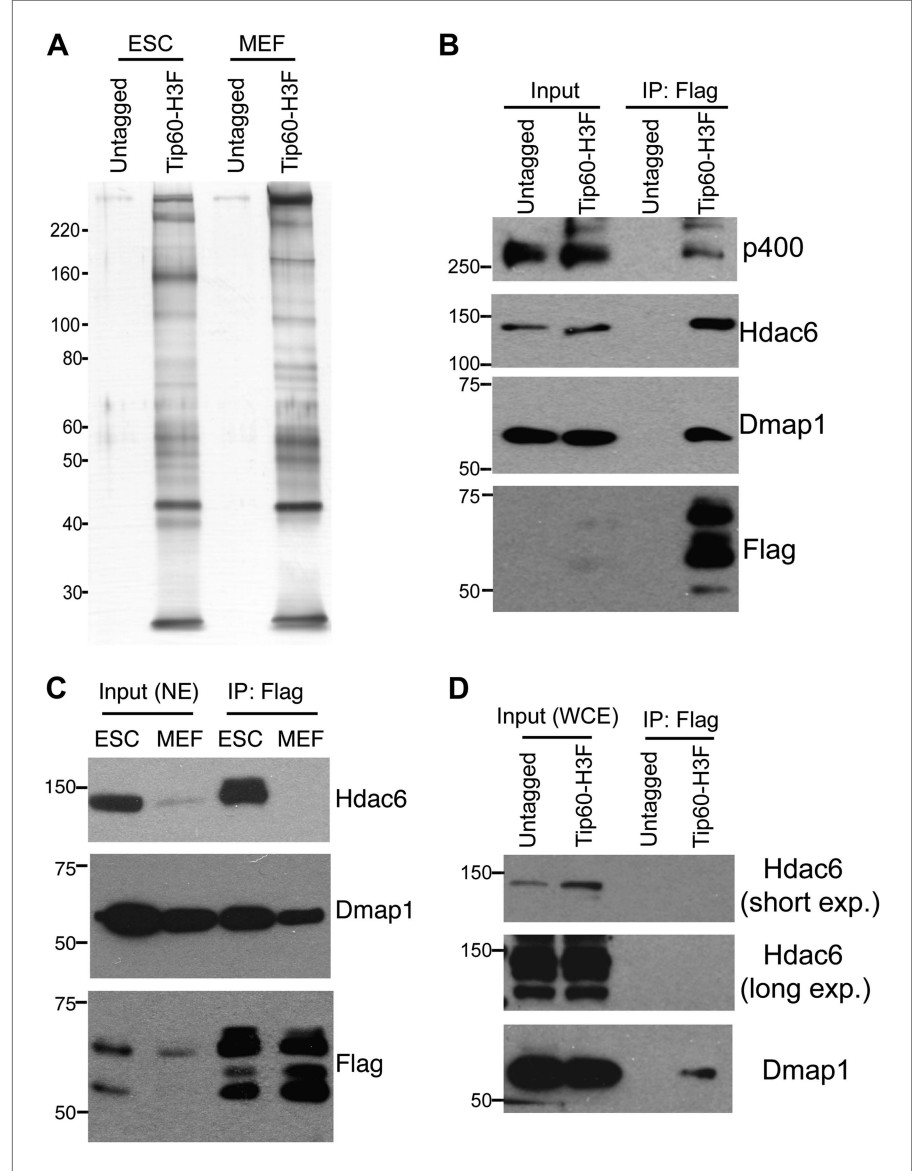

**Figure 1**. Identification of Tip60-p400-interacting proteins in ESCs. (**A**) Silver stained gel of purified Tip60 complex from Tip60-H3F ESCs and MEFs, along with untagged control cells. (**B**) Validation of Hdac6 interaction. Western blots for Hdac6, p400, and Dmap1 following immunoprecipitation with anti-FLAG antibody from nuclear extracts derived from the indicated ESC lines. (**C**) Tip60 complexes purified from Tip60-H3F ESC and MEF nuclear extracts were subjected to Western blotting for Hdac6, Dmap1, and FLAG. (**D**) Tip60-H3F was immunoprecipitated from MEF whole cell extracts as above and blotted for the indicated proteins.

The following figure supplements are available for figure 1:

**Figure supplement 1**. Targeting of H3F tag to C-terminus of endogenous *Tip60* gene.

**Figure supplement 2**. Reciprocal co-immunoprecipitation of Tip60 and Hdac6.

**Figure supplement 3**. Hdac6 does not interact with RNA Polymerase II in ESCs.

**Figure supplement 4**. Hdac6 interaction with Tip60-p400 is independent of DNA.

**Figure supplement 5**. Hdac6 interaction with Tip60-p400 is resistant to high salt concentrations.

**Table 1.** Proteins co-purifying with Tip60-H3F in ESCs

| Name | Description | # of peptides | MW (kD) | Peptides/ MW | Gel slices |
|---|---|---|---|---|---|
| Stk38 | Serine/threonine-protein kinase 38 | 78 | 54 | 1.44 | 7,14,15,16,17 |
| **Ruvbl1** | RuvB-like 1 | 42 | 50 | 0.84 | 7,8,13,15,16,17 |
| **Ruvbl2** | RuvB-like 2 | 38 | 51 | 0.75 | 15,16,17 |
| Acta1 or other iso | Actin | 21 | 42 | 0.50 | 18 |
| Sun2 | Protein unc-84 homolog B | 36 | 82 | 0.44 | 11,12,13 |
| Hdac6 | Histone deacetylase 6 | 38 | 126 | 0.30 | 1,2,3,4,5,6,7,8 |
| **Kat5** | Histone acetyltransferase KAT5 | 17 | 59 | 0.29 | 13,14,15 |
| Actb | Actin, cytoplasmic 1 | 12 | 42 | 0.29 | 5,8,9 |
| **Trrap** | Transformation/transcription domain-associated protein | 72 | 292 | 0.25 | 1,2 |
| **Epc1** | Enhancer of polycomb homolog 1 | 22 | 90 | 0.24 | 9,10 |
| **Brd8** | Bromodomain-containing protein 8 | 24 | 103 | 0.23 | 6,7 |
| **Yeats4** | YEATS domain-containing protein 4 | 6 | 27 | 0.22 | 20 |
| **Epc2** | Enhancer of polycomb homolog 2 | 20 | 91 | 0.22 | 10,11 |
| H2B | Histone H2B | 3 | 14 | 0.21 | 21 |
| **Ing3** | Inhibitor of growth protein 3 | 10 | 47 | 0.21 | 16,17 |
| **Ep400** | E1A-binding protein p400 | 70 | 337 | 0.21 | 1,2 |
| **Dmap1** | DNA methyltransferase 1-associated protein 1 | 10 | 53 | 0.19 | 14,15 |
| Hspa8 | Heat shock cognate 71 kDa protein | 13 | 69 | 0.19 | 13 |
| Lima1 | LIM domain and actin-binding protein 1 | 15 | 84 | 0.18 | 9,10 |
| **Vps72** | Vacuolar protein sorting-associated protein 72 homolog | 7 | 41 | 0.17 | 16,17 |
| **Actl6a** | Actin-like protein 6A | 8 | 47 | 0.17 | 17,18 |
| Actg1 | Actin, cytoplasmic 1 | 6 | 42 | 0.14 | 19 |
| H2afv | Histone H2A.V | 2 | 14 | 0.14 | 22 |
| **Meaf6** | Chromatin modification-related protein MEAF6 | 3 | 22 | 0.14 | 20 |
| Mbtd1 | MBT domain-containing protein 1 | 8 | 71 | 0.11 | 13,14 |
| Rps18 | 40S ribosomal protein S18 | 2 | 18 | 0.11 | 21 |
| Tubb5 | Tubulin beta-5 chain | 5 | 50 | 0.10 | 16 |
| Tuba1a or other iso | Tubulin alpha chain | 5 | 50 | 0.10 | 16 |
| Trim28 | Transcription intermediary factor 1-beta | 8 | 89 | 0.09 | 9,10 |
| **Morf4l2** | Mortality factor 4-like protein 2 | 3 | 34 | 0.09 | 19 |
| **Mrgbp** | MRG-binding protein | 2 | 24 | 0.08 | 20 |
| Rangap1 | Ran GTPase-activating protein 1 | 5 | 64 | 0.08 | 12 |
| Spna2 | Spectrin alpha chain, brain | 18 | 285 | 0.06 | 3 |
| Setx | Probable helicase senataxin | 18 | 298 | 0.06 | 2,3 |
| Sf3b1 | Splicing factor 3B subunit 1 | 3 | 54 | 0.06 | 6 |
| Sfpq | Splicing factor, proline- and glutamine-rich | 4 | 75 | 0.05 | 10 |
| Rab5c | Ras-related protein Rab-5C | 1 | 23 | 0.04 | 20 |
| Lrrfip2 | Leucine-rich repeat flightless-interacting protein 2 | 2 | 47 | 0.04 | 16 |
| Stat2 | Signal transducer and activator of transcription 2 | 2 | 50 | 0.04 | 8 |
| Nono | Non-POU domain-containing octamer-binding protein | 2 | 55 | 0.04 | 15 |

*Table 1. Continued on next page*

*Table 1. Continued*

| Name | Description | # of peptides | MW (kD) | Peptides/ MW | Gel slices |
|---|---|---|---|---|---|
| Tpr | Nucleoprotein TPR | 6 | 274 | 0.02 | 3 |
| Hnrnpf | Heterogeneous nuclear ribonucleoprotein F | 1 | 46 | 0.02 | 17 |
| Spnb2 | Spectrin beta chain, brain 1 | 5 | 274 | 0.02 | 3 |
| Flna | Filamin-A | 4 | 281 | 0.01 | 3 |
| Flii | Protein flightless-1 homolog | 2 | 145 | 0.01 | 6 |
| Hdx | Highly divergent homeobox | 1 | 77 | 0.01 | 11 |
| *Morf4l1* | Mortality factor 4-like protein 1 | 0 | 41 | 0 | N/A |

Proteins in bold represent known Tip60-p400 subunits found in Tip60-H3F purification from ESCs. The protein in bold italic represents the known Tip60-p400 subunit not found in purification from ESCs.

Previously, both Hdac6 and Tip60 were separately found to interact with RNA Polymerase II (Pol II) in human CD4[+] T-cells (*Wang et al., 2009*), raising the possibility that the Hdac6-Tip60 interaction we observed in ESCs might be mediated by Pol II. However, there were no peptides corresponding to Pol II in our LC-MS/MS data, and we could not detect Pol II in Tip60-H3F immunoprecipitates (*Figure 1— figure supplement 3*), arguing against this explanation. Furthermore, the Tip60-Hdac6 interaction was independent of DNA and resistant to high salt (*Figure 1—figure supplement 4*, *Figure 1—figure supplement 5*), verifying that Hdac6 is a stable interaction partner within Tip60-p400 complex. Finally, we tested whether Hdac6 interacts with Tip60-p400 complex in a differentiated cell type, MEFs. Unlike ESCs, Tip60-H3F immunoprecipitated from MEF nuclear extracts or whole cell lysates did not pull down Hdac6 (*Figure 1C–D*), consistent with the possibility that Hdac6 interacts with Tip60-p400 complex in only a subset of cell types.

The reported cytoplasmic localization of Hdac6 in multiple types of cells (*Verdel et al., 2000*; *Hubbert et al., 2002*; *Kawaguchi et al., 2003*; *Valenzuela-Fernández et al., 2008*) raised the question of whether its interaction with Tip60-p400 complex was physiologically relevant. To address this issue, we first confirmed that Hdac6 exhibited significant nuclear localization in ESCs, in contrast to MEFs, in which Hdac6 was mainly cytoplasmic (*Figure 2A*). Next, we prepared cytoplasmic and nuclear protein fractions to examine the cellular localization of Hdac6 in ESCs and several adult cell types by Western blotting. Interestingly, while differentiated cells (MEFs, whole brain) exhibited the reported cytoplasmic sequestration, high levels of Hdac6 in undifferentiated cells, including ESCs, NSCs, and hematopoietic stem and progenitor cells (HPCs), were found in the nucleus (*Figure 2B*). Consistent with these data, differentiation of either ESCs or NSCs caused a dramatic decrease in nuclear Hdac6, accompanied by increased Hdac6 within the cytoplasm (*Figure 2C–D*). To test whether nuclear localization of Hdac6 promoted its interaction with Tip60-p400 complex, we immunoprecipitated Tip60 from NSCs isolated from *Tip60-H3F* knock-in mice. Indeed, Hdac6 was present in Tip60-H3F immunoprecipitates from NSC nuclear extracts (*Figure 2E*). These data suggest that the interaction of Hdac6 with Tip60-p400 in undifferentiated cells is lost during the course of differentiation of some types of stem cells, due to nuclear exclusion of Hdac6. However, cytoplasmic sequestration in differentiated cells cannot be the sole factor preventing Hdac6 from associating with Tip60-p400, since we did not observe Hdac6 within Tip60-H3F immunoprecipitates from MEF whole cell lysates (in which nuclear and cytoplasmic proteins are mixed; *Figure 1D*). Together, these data show that Hdac6 exhibits significant nuclear localization in some types of embryonic and adult stem and progenitor cells, where it associates with Tip60-p400 complex. Furthermore, stem cell differentiation promotes re-localization of Hdac6 to the cytoplasm, where it is sequestered away from Tip60-p400.

## Hdac6 interacts with Tip60-p400 in some cancer cells

While Hdac6 localizes to the cytoplasm of normal somatic cells, nuclear Hdac6 has been observed in some cancers, in particular within tumors that are poorly differentiated (*Subramanian et al., 2011*; *Riolo et al., 2012*). We therefore tested several human cancer cell lines and found that Hdac6 exhibited significant nuclear localization in each of the cell lines (*Figure 2F*). We further tested whether Tip60-p400 interacts with Hdac6 in cancer cells, by transfecting epitope-tagged *Tip60* and *Hdac6* constructs into

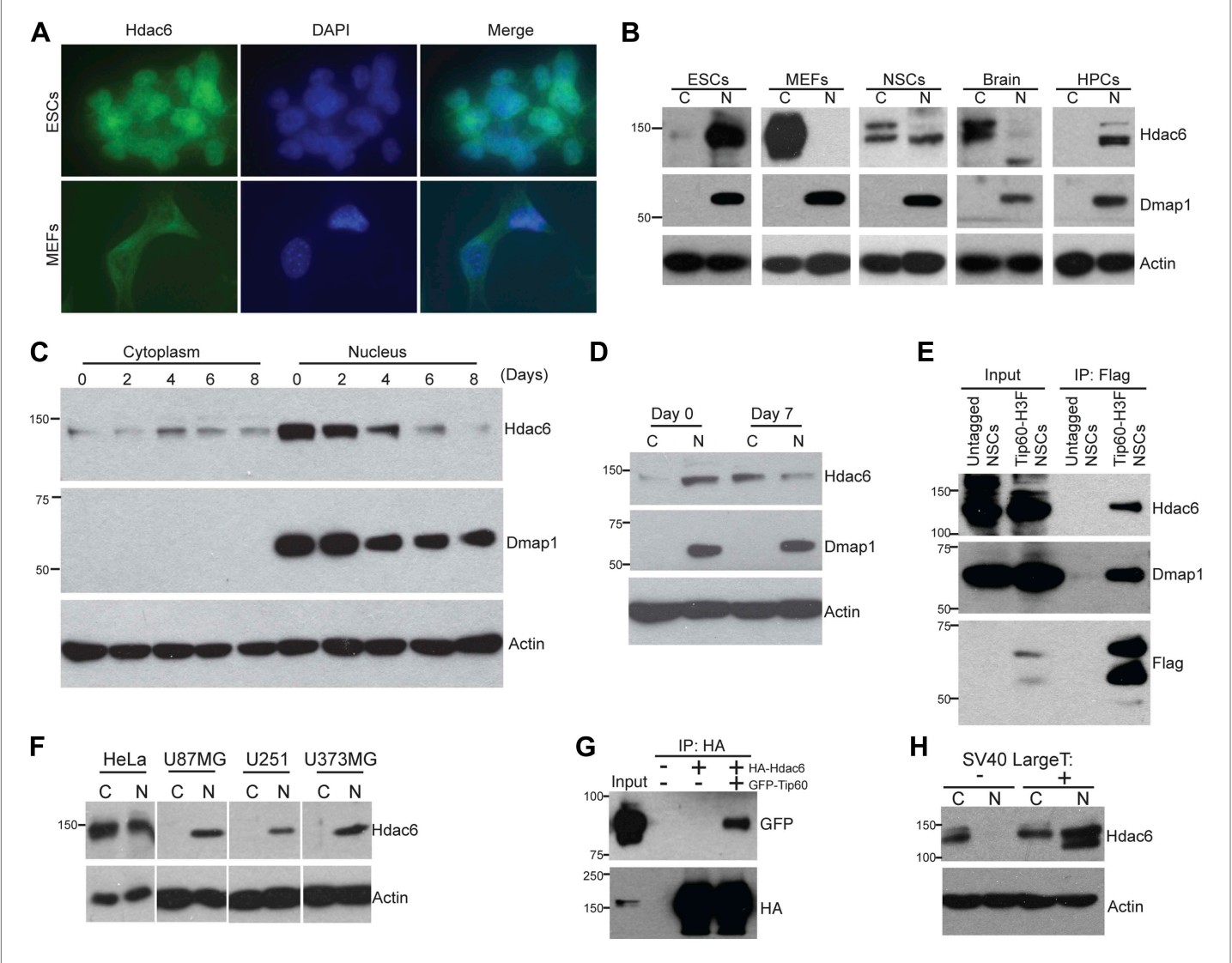

**Figure 2**. Hdac6 is partially nuclear in multiple types of undifferentiated cells and interacts with Tip60-p400 in NSCs and cancer cell lines. (**A**) ESCs (top) or MEFs (bottom) were subjected to immunofluorescence using an antibody recognizing Hdac6. DAPI staining is shown to identify nuclei. (**B**) High levels of nuclear Hdac6 in stem cells but not differentiated cells. Western blots of Hdac6 and known Tip60-p400 subunit Dmap1 in cytoplasmic (C) and nuclear (N) fractions of indicated cells are shown, with β-actin serving as a loading control. (**C**) Hdac6 relocalizes to the cytoplasm during ESC differentiation. Cytoplasmic (C) and nuclear (N) fractions from ESCs differentiated for the indicated number of days were Western blotted for the indicated proteins. (**D**) Hdac6 relocalizes to the cytoplasm during NSC differentiation. Cytoplasmic (C) and nuclear (N) fractions from undifferentiated NSCs (day 0) or NSCs differentiated for 7 days were Western blotted for the indicated proteins. (**E**) Hdac6 interacts with Tip60-p400 in NSCs. Shown are Western blots for the proteins indicated of input or Tip60-p400 complex immunoprecipitated from NSCs. (**F**) Hdac6 is nuclear localized in cancer cell lines. Cells were fractionated and Western blotted as in (**B**). (**G**) Hdac6 interacts with Tip60-p400 in a cancer cell line. 293.T cells were transfected with the indicated constructs. Nuclear extracts were prepared, subjected to immunoprecipitation with an anti-HA antibody, and Western blotted as indicated. (**H**) Re-localization of Hdac6 to the nucleus is an early event during transformation. MEFs were infected with pBABE-puro retrovirus expressing SV40 large T antigen or empty vector, harvested after 5 days (including 3 days of selection), and fractionated as in (**B**). Western blots are shown for the proteins indicated.

293.T cells (to facilitate immunoprecipitation), and found that Tip60 co-precipitated with Hdac6 (*Figure 2G*), suggesting that, as with ESCs and NSCs, Hdac6 functions within Tip60-p400 complex in some cancer cells. Interestingly, loss-of-function of either *Hdac6* or genes encoding Tip60-p400 subunits within cancer cell lines has previously been shown to elicit defects in anchorage-independent growth and hypersensitivity to DNA damaging agents (*Feng et al., 2003*; *Lee et al., 2008*; *Wang et al., 2012*), supporting this hypothesis. Finally, we asked whether re-localization of Hdac6 to the nucleus in

cancer cells is an early event during transformation, at a time when it might be more likely to contribute to cancer progression. We found that shortly after introduction of SV40 large T antigen into MEFs, a large fraction of Hdac6 re-localized to the nucleus (*Figure 2H*), suggesting that nuclear localization of Hdac6 is an early event during transformation. Together, these data suggest that Hdac6 functions within Tip60-p400 complex in some types of cancer, and support the idea that stem cells and some cancer cells share several common phenotypes and regulatory pathways.

## Hdac6 is necessary for normal regulation of most Tip60-p400 target genes

To test whether Hdac6 is necessary for gene regulation by Tip60-p400 complex in ESCs, we used DNA microarrays to examine the changes in mRNA levels upon *Tip60* or *Hdac6* KD. We observed highly correlated gene expression profiles in ESCs knocked down individually for *Tip60* and *Hdac6* (R = 0.63), suggesting a significant overlap in their sets of target genes, although *Hdac6* KD generally had weaker effects on expression of common targets (*Figure 3A–B*). Next, we performed unsupervised hierarchical clustering of mRNA expression data comparing ESCs depleted of Tip60, Hdac6, or both to control KD ESCs. We observed four main clusters of genes differentially expressed in *Tip60* KD cells: genes upregulated upon *Tip60* KD but unaffected by *Hdac6* KD (*Figure 3C*, cluster 1), genes upregulated in both single KDs (cluster 2), genes downregulated in *Tip60* KD cells but unaffected by *Hdac6* KD (cluster 3), and genes downregulated in both single KDs (cluster 4). Genes downregulated upon *Tip60* KD were significantly overrepresented (relative to the number expected by chance) among Hdac6-independent Tip60 targets, while genes upregulated upon *Tip60* KD were significantly overrepresented among Hdac6-dependent target genes (*Figure 3D*), suggesting that Tip60-dependent repression in ESCs usually requires Hdac6. Double KD of *Tip60* and *Hdac6* was nearly identical to the *Tip60* single KD, consistent with the model that Hdac6 functions within Tip60-p400 complex (*Figure 3C*). We next tested individual genes from each cluster by RT-qPCR, which generally confirmed the microarray results (*Figure 3E*). We further validated these results using two *Hdac6* mutant ESC lines: one that harbors a hypomorphic allele that expresses reduced Hdac6 levels (*Hdac6reduced*), and one that contains an *Hdac6* deletion (*Hdac6null*; *Figure 3—figure supplement 1*, *Figure 3—figure supplement 2*). Both lines exhibited misregulation of Tip60/Hdac6 target genes similar to that observed upon *Hdac6* KD, with greater effects usually observed in the more severe *Hdac6null* line (*Figure 3—figure supplement 3*). Finally, we examined which classes of genes were enriched in each regulatory cluster. While Hdac6-independent targets of Tip60-p400 were enriched for genes involved in cellular growth, homeostasis, and the cell cycle (*Figure 3—figure supplement 4*, clusters 1 and 3), the majority of Hdac6-dependent Tip60-p400 target genes were differentiation-induced genes (*Figure 3—figure supplement 4*, cluster 2), consistent with the idea that Hdac6 is broadly important for Tip60-p400-dependent repression of developmental genes, although it also plays a smaller role in the activation of some Tip60-dependent proliferation genes (*Figure 3—figure supplement 4*, cluster 4).

## Hdac6 binding overlaps one of two peaks of Tip60 enrichment at common target promoters

Despite overlapping roles in gene regulation, the function of Hdac6 within Tip60-p400 complex remained unclear. Hdac6 is not thought to bind chromatin or regulate gene expression in most cell types (*Verdel et al., 2000*; *Hubbert et al., 2002*; *Haggarty et al., 2003*), therefore we considered the possibility that Hdac6 modulates Tip60-p400 function prior to chromatin binding by the complex. To determine whether Hdac6 associates with chromatin-bound Tip60-p400, we tested whether Hdac6 co-localizes with Tip60-p400 on chromatin. To this end, we examined the genome-wide distributions of Tip60 and Hdac6 in ESCs using chromatin immunoprecipitation followed by deep sequencing of the precipitated DNA (ChIP-seq). To facilitate these analyses, we generated an ESC line in which the H3F tag utilized above was fused to the C-terminus of Hdac6 at the endogenous *Hdac6* locus (*Figure 4—figure supplement 1*). This line allowed us to directly compare the genomic binding profiles of Tip60 and Hdac6 with untagged control cells using the same antibody, thereby eliminating differences in background.

We found that Tip60 was enriched at the 5′ ends of many genes in ESCs, with two peaks of binding flanking the promoter regions of most targets, one at approximately 400 base pairs upstream and another at approximately 100 base pairs downstream of the TSS (*Figure 4A–B*). This two peak pattern of Tip60 binding and the gene set bound by Tip60 were very similar to previous mapping data examining the distribution of the p400 subunit of Tip60-p400 complex in ESCs (*Fazzio et al., 2008a*). Interestingly, we found that while Hdac6 was also enriched at p400- (*Figure 4A–B*) and Tip60-target genes (*Figure 4C*)

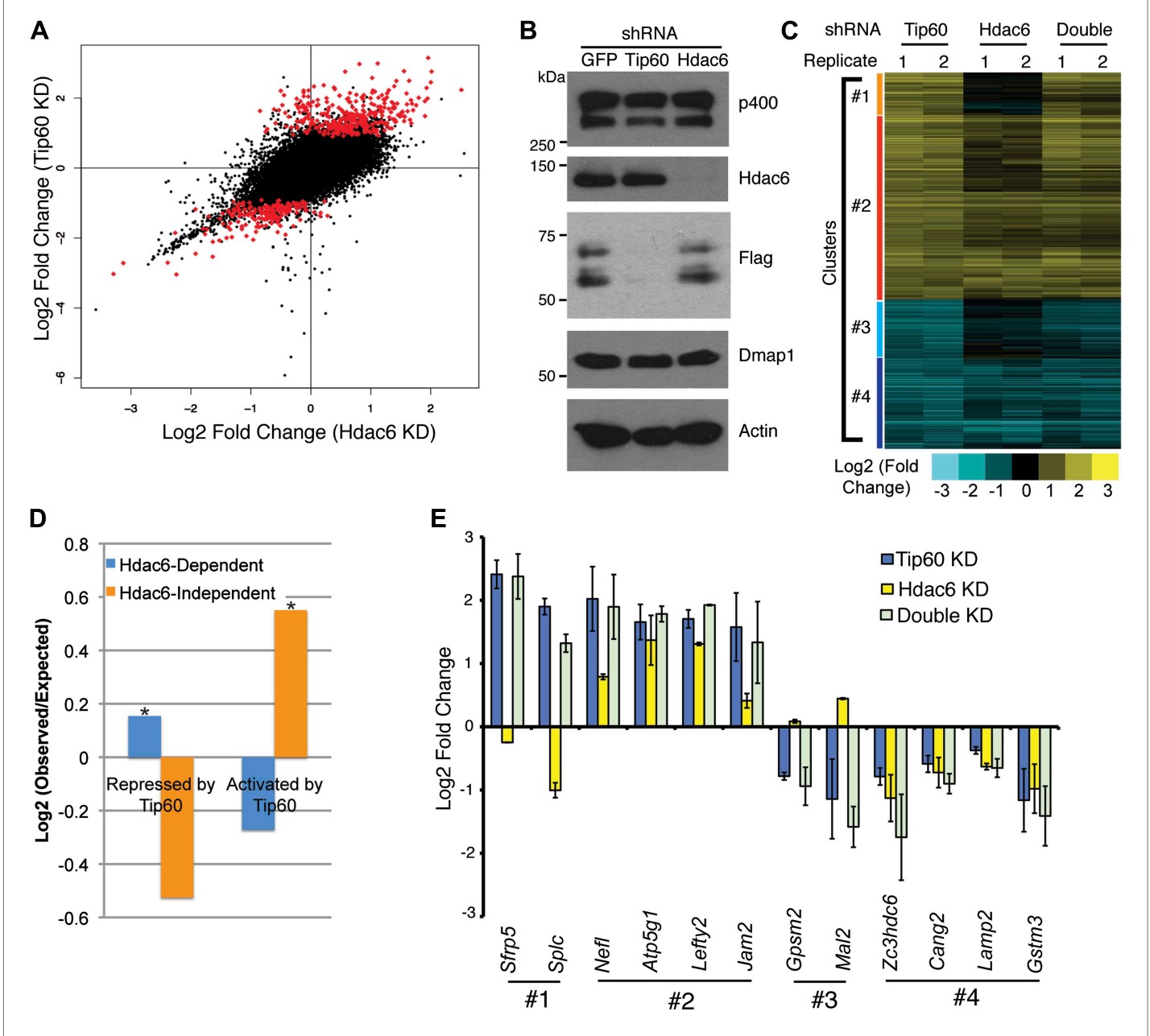

**Figure 3**. Overlapping effects of *Hdac6* KD and *Tip60* KD on gene expression in ESCs. (**A**) Scatter plot of gene expression (Log2 (fold change) relative to control KD ESCs) upon *Tip60* KD relative to *Hdac6* KD. Genes misregulated upon *Tip60* KD (adjusted p<0.1) are shown in red. (**B**) Western blot showing the levels of p400, Dmap1, Hdac6 and Tip60 (FLAG) upon *Tip60* or *Hdac6* KD. β-actin is shown as a loading control. (**C**) Unsupervised hierarchical clustering of genes misregulated (adjusted p<0.1) upon *Tip60* KD. Up-regulated genes are indicated in yellow and downregulated genes are indicated in blue. The first cluster (#1) includes 200 genes that were upregulated upon *Tip60* KD but not *Hdac6* KD, the second cluster (#2) includes 867 genes upregulated in both *Tip60* KD and *Hdac6* KD ESCs, the third cluster (#3) includes 277 genes that were downregulated only in *Tip60* KD cells and the forth cluster (#4) includes 424 genes that were downregulated in both *Tip60* KD and *Hdac6* KD ESCs. (**D**) Hdac6-dependent target genes are biased toward genes repressed by Tip60. Genes repressed or activated by Tip60 were split based on their Hdac6-dependence, and each group was plotted as the Log2 (ratio) of genes observed in each group relative to the expected number of genes if Hdac6-dependence was randomly distributed. Asterisk indicates statistically significant enrichment (p<10$^{-20}$). (**E**) Validation of microarray datasets. The expression levels of genes from each cluster were measured by RT-qPCR in the indicated KDs and expressed as Log2 (fold change) values relative to control (*GFP*) KD ESCs after normalization. Data shown are mean ± SD of three technical replicates from one representative experiment of two biological replicates performed.

*Figure 3. Continued on next page*

*Figure 3. Continued*

The following figure supplements are available for figure 3:

**Figure supplement 1**. *Hdac6* mutant ESCs.

**Figure supplement 2**. Hypomorphic and null *Hdac6* mutant ESCs.

**Figure supplement 3**. *Hdac6* mutant ESCs exhibit alterations in Tip60-target gene expression consistent with KD phenotypes.

**Figure supplement 4**. GO-term enrichment of gene clusters.

near their TSSs, its pattern of binding at each gene was somewhat different, forming one peak of enrichment at approximately 100 base pairs downstream of the TSS that overlapped with the downstream Tip60 peak (*Figure 4A–B*). Like p400 (*Fazzio et al., 2008a*), we found that Hdac6 was enriched at genes marked by H3K4me3, including bivalent genes also marked by H3K27me3 (*Figure 4—figure supplement 2*). Furthermore, we found that, on average, Hdac6 and Tip60 were both enriched to significantly higher levels at the promoter regions of genes that are misregulated upon *Hdac6* or *Tip60* KD compared to the genes unaffected by KD of these factors (*Figure 4D*), suggesting that many of these genes are direct targets. Interestingly, while the levels of Hdac6 binding were elevated at genes from clusters 2 and 4 (*Figure 3C*), whose expression levels are regulated by Hdac6, they were also elevated at cluster 3 genes, whose expression levels are Tip60-dependent but Hdac6-independent (*Figure 4E*), suggesting that Tip60-p400 acts independently of Hdac6 to activate these genes. Together, these data show that Hdac6 binds in an asymmetrical pattern with respect to the transcription start site at its genomic targets. In addition, the overlap between Hdac6 and one of the two peaks of Tip60-p400 binding is consistent with a model in which Hdac6 functions within chromatin-bound Tip60-p400 complex to regulate common target genes in ESCs. Consistent with our finding that Hdac6 does not interact with Tip60-p400 in differentiated cells, we found that Hdac6-dependent Tip60-target genes in ESCs were not bound by Tip60 in MEFs (*Figure 4—figure supplement 3*).

## The catalytic domains of Hdac6 are necessary for interaction with Tip60-p400 complex, but not for deacetylation of histones at target promoters

We next tested several possible models by which Hdac6 might function within Tip60-p400 complex to regulate gene expression. First, we considered the possibility that Hdac6 was necessary for complex formation or stability. To test this model, we purified Tip60-H3F in control and *Hdac6* KD ESCs, but found that the composition of Tip60-p400 complex was similar in the presence or absence of Hdac6 (*Figure 5—figure supplement 1*), arguing against this explanation. Alternatively, Hdac6 might be necessary for localization of Tip60-p400 to its target genes in ESCs. Finally, Tip60-p400 may recruit Hdac6 to its target genes, where it regulates gene expression by deacetylating histones. To distinguish between these latter two possibilities, we performed a series of experiments. First, we tested whether the treatment of ESCs with tubastatin A (TubA), a chemical inhibitor that prevents Hdac6-dependent deacetlyation by binding within the catalytic channel of Hdac6's deacetylase domains (*Butler et al., 2010*), had similar effects on gene expression as *Hdac6* KD. Like *Hdac6* KD or *Tip60* KD ESCs, most target genes were de-repressed upon TubA treatment (*Figure 5A*), consistent with the possibility that histone deacetylation by Hdac6 was necessary for repression of its gene targets. TubA treatment did not affect expression or nuclear localization of Hdac6 (*Figure 5—figure supplement 2*), suggesting that it acts directly on the Hdac6 deacetylase domains.

Next, we tested whether *Hdac6* KD resulted in increased acetylation of histone tails at Hdac6 target promoters by examining acetylation of the N-terminal tails of histones H4 and H2A, two major targets of the Tip60 acetyltransferase activity (*Yamamoto and Horikoshi, 1997*; *Kimura and Horikoshi, 1998*; *Yan et al., 2000*; *Doyon et al., 2004*; *Altaf et al., 2010*). ChIP-qPCRs using antibodies recognizing histone H4 acetylated at all four N-terminal lysines or acetylated lysine-5 of histone H2A were performed in control, *Tip60* KD, and *Hdac6* KD ESCs. Surprisingly, like *Tip60* KD, *Hdac6* KD resulted in a decrease in both H4 and H2A acetylation at most shared targets of Tip60 and Hdac6 that were examined (*Figure 5B*, *Figure 5—figure supplement 3*). These findings show that Hdac6 does not silence differentiation genes

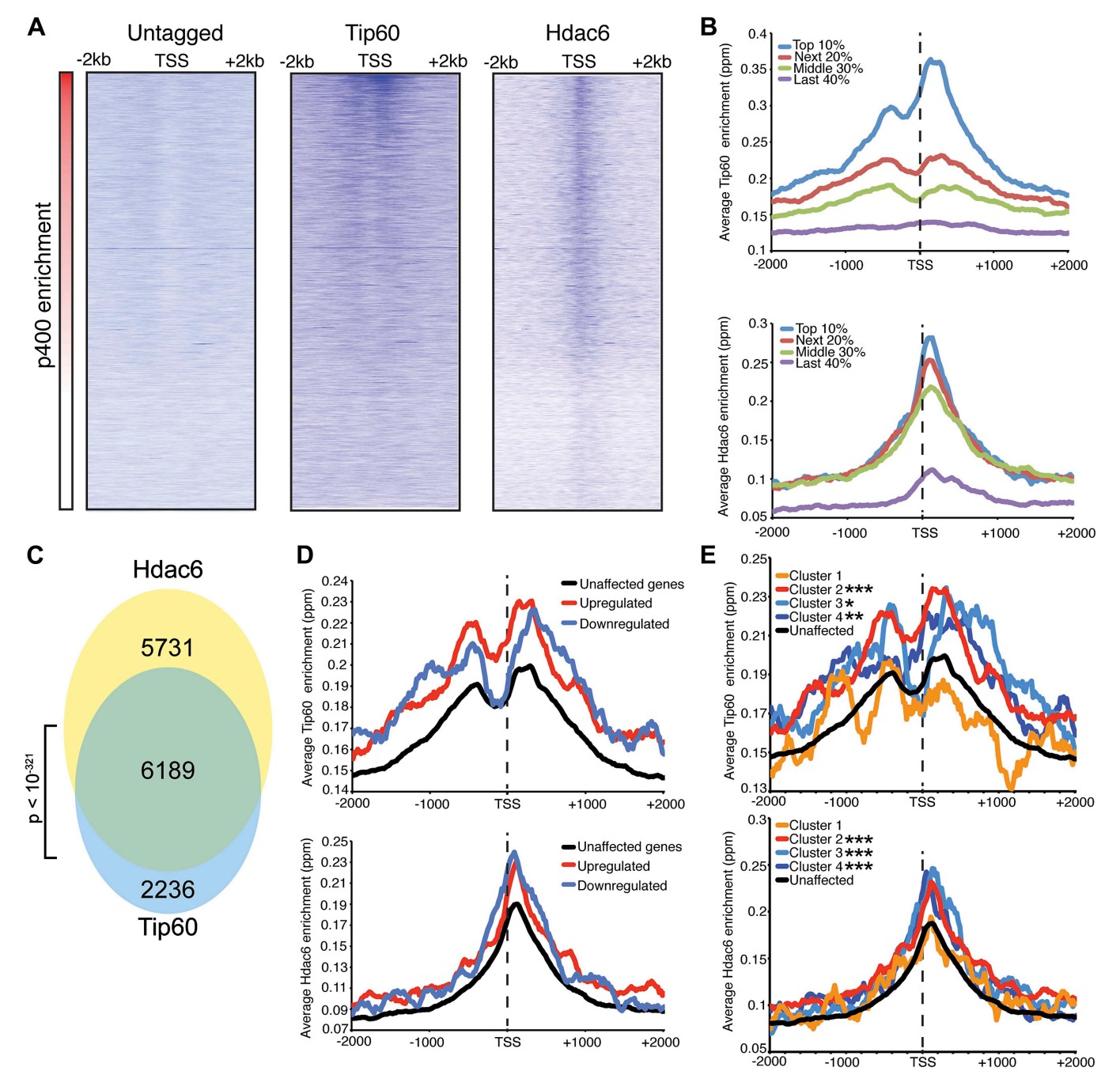

**Figure 4**. Tip60 and Hdac6 co-localize on chromatin. (**A**) Heat map representation of ChIP-seq data for H3F-tagged Tip60 and Hdac6 comprising the 2 kb surrounding the transcriptional start sites (TSS) of 10,507 genes for which published p400 ChIP-chip data (***Fazzio et al., 2008a***) (p400 enrichment) were available. ChIP-seq data from E14 (Untagged) cells is shown as a control. All panels are sorted by decreasing p400 binding for the 1 kb surrounding the TSS, ranging from high levels of p400 binding (red) to genes unbound by p400 (white). (**B**) Tip60 and Hdac6 binding correlate with p400 binding. Genes in the p400 ChIP-chip dataset were grouped by the intensity of p400 enrichment: The groups of genes exhibiting the top 10% of p400 enrichment (top 10%), the 11th–30th percentile (next 20%), the 31st–60th percentile (middle 30%) and the rest of genes in dataset (last 40%). Upper panel: averaged Tip60 enrichment for groups of genes at each level of p400 binding are shown relative to the TSS. Lower panel: averaged Hdac6 binding data for genes in the same groups. (**C**) Correlation of Tip60 and Hdac6 binding. Shown is a Venn diagram delineating the overlap between the gene sets bound by Tip60 and Hdac6. The p-value was calculated by summing the hypergeometric probabilities of Tip60/Hdac6 overlap below the number observed and subtracting from one. (**D**) Hdac6 and Tip60 are enriched at genes regulated by these factors. Upper panel: Tip60 binding segregated by genes that are

*Figure 4. Continued on next page*

*Figure 4. Continued*

upregulated, downregulated, or unchanged by *Tip60* KD. Bottom panel: Hdac6 binding at genes segregated as in upper panel. (**E**) Tip60 and Hdac6 are enriched at genes within clusters 2, 3, and 4. Tip60 (upper panel) and Hdac6 (bottom panel) binding data are shown for genes segregated by cluster, as in *Figure 2C*. Asterisks mark clusters exhibiting statistically significantly higher promoter-proximal (−500 to +500) binding of indicated factor than does the set of genes not regulated by Tip60 and Hdac6: *(p<0.05); **(p<0.01); ***(p<10-5).

The following figure supplements are available for figure 4:

**Figure supplement 1**. Targeting of the H3F tag to the C-terminus of the endogenous Hdac6 gene.

**Figure supplement 2**. Hdac6 is enriched at genes marked by H3K4me3.

**Figure supplement 3**. Hdac6-dependent Tip60 targets in ESCs are not bound by Tip60 in MEFs.

by counteracting Tip60-dependent histone acetylation, despite the fact that treatment with TubA resulted in gene expression changes similar to *Hdac6* KD. *Hdac6* KD had no significant effect on bulk histone H4 acetylation, although tubulin acetylation was strongly enhanced by *Hdac6* KD (*Figure 5—figure supplement 4*), consistent with a model in which Hdac6 is required for Tip60 function only at common target promoters.

A few subunits of Tip60-p400 complex are known to be acetylated (*Choudhary et al., 2009*), raising the possibility that Hdac6 regulates Tip60-p400 function by deacetylating the complex, which then leads to altered biochemical activity or chromatin binding by the complex. We tested this possibility by Western blotting Tip60-p400 complex purified from control or *Hdac6* KD ESCs with an antibody specific for acetyl-lysine, but we did not observe detectable acetylation of any subunit in either KD (*Figure 5—figure supplement 5*). Additionally, we performed mass spectrometry on p400, the most highly acetylated subunit in Tip60-p400 complex (*Choudhary et al., 2009*), purified from control or Hdac6 KD ESCs, but did not observe any notable differences in acetylation levels (data not shown). Thus, while the loss of *Hdac6* function does not lead to an observable increase in acetylation of Tip60-p400 complex or histone tails, it does lead to de-repression of Tip60-p400 target genes. Next, we considered the possibility that Hdac6 binds Tip60-p400 through one or both of its deacetylase domains. In this scenario, despite the fact that Hdac6 does not appear to deacetylate Tip60-p400, TubA could cause de-repression of Tip60- and p400-target genes simply by preventing Hdac6 from binding Tip60-p400. We tested this possibility by immunoprecipitating Tip60-p400 complex from *Tip60-H3F* ESCs treated with either vehicle alone or TubA and determining whether Hdac6 co-precipitates with Tip60. Interestingly, TubA treatment completely abolished Hdac6 association with Tip60-p400 (*Figure 5C*).

We reasoned that if TubA directly inhibited binding of Hdac6 to Tip60-p400, then addition of the drug to purified Hdac6-containing Tip60-p400 complex should disrupt this interaction. In contrast, if Hdac6 deacetylates some unknown protein, which then activates it to bridge the interaction of Hdac6 with Tip60-p400, the treatment of cells with TubA should disrupt the Hdac6 interaction with Tip60-p400, while in vitro TubA treatment of Tip60-p400 complex should not disrupt the Hdac6 interaction. To distinguish between these two possibilities, we immunoprecipitated Tip60-p400 from untreated *Tip60-H3F* ESCs and, after washing unbound proteins away from beads, subjected them to five additional washes with or without TubA. After harvesting the protein eluted in the TubA washes or remaining bound to beads, we examined the distribution of Hdac6 by Western blotting. Interestingly, we found that the TubA washes removed a large fraction of Hdac6 from bead-bound Tip60-p400 complex (*Figure 5D*), suggesting that TubA directly disrupts the interaction of Hdac6 with Tip60-p400, rather than preventing deacetylation of histones or other proteins. Consistent with these findings, we found that mutation of the Hdac6 deacetylase domains had the same effect as TubA treatment: mutation of deacetylase domain 1 partially disrupted Hdac6's interaction with Tip60-p400 in ESCs, while mutation of both deacetylase domains abolished this interaction (*Figure 5E*). Together, these data indicate that Hdac6 interacts with Tip60-p400 via its deacetylase domains and that TubA directly disrupts this interaction.

## Hdac6 is necessary for maximal Tip60 and p400 recruitment

The finding that Hdac6 does not appear to deacetylate histones or Tip60-p400 subunits, but that its deacetylase domains are necessary for interaction with Tip60-p400 complex, was consistent with a

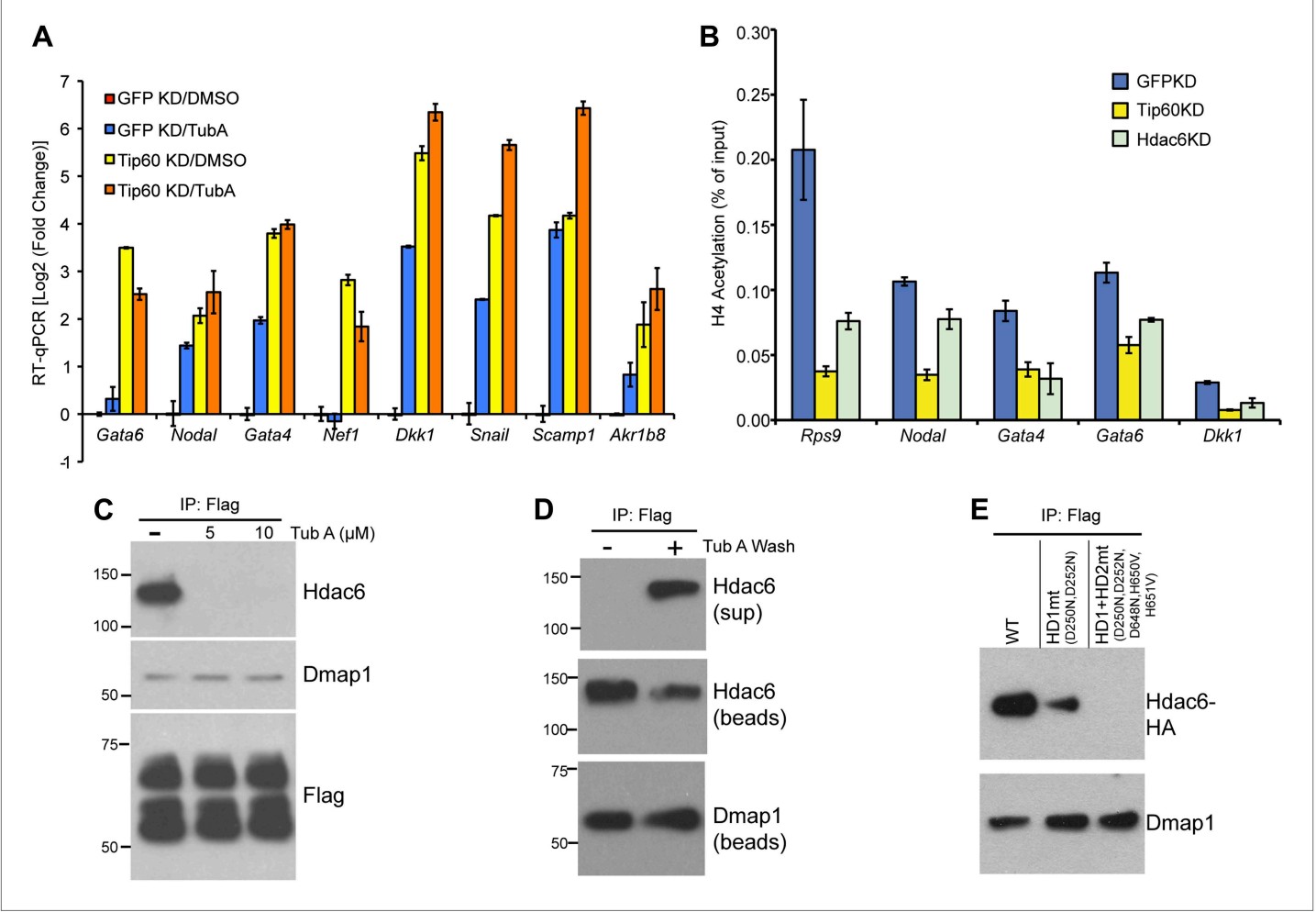

**Figure 5**. The Hdac6 deacetylase domains are necessary for Tip60-p400 binding but do not reverse histone acetylation catalyzed by Tip60. (**A**) Treatment of ESCs with Hdac6 catalytic domain inhibitor Tubastatin A causes de-repression of Tip60-p400 target genes. RT-qPCRs for indicated genes are shown for *GFP* KD or *Tip60* KD ESCs treated with either Tubastatin A (TubA) or DMSO vehicle. Data are expressed as Log2 (fold change) values relative to DMSO treated *GFP* KD ESCs after normalization. Shown are the mean ± SD values of three technical replicates from one representative experiment of two biological replicates performed. (**B**) H4 acetylation levels for several common Tip60/Hdac6 target genes in *Tip60* KD and *Hdac6* KD ESCs were measured by ChIP-qPCR, using an antibody specific for tetra-acetylated histone H4. H4 acetylation levels in cells knocked down as indicated are expressed as a fraction of the input. Shown are the mean ± SD values of three technical replicates from one representative experiment of two biological replicates performed. (**C**) ESCs were treated overnight with the indicated amounts of TubA in their growth medium, Tip60-H3F was immunoprecipitated as above, and co-immunoprecipitating proteins were examined by Western blotting. (**D**) Tip60-H3F was immunoprecipitated from ESCs grown under normal conditions, and the beads were washed in buffer with or without TubA. The Hdac6 eluted in the TubA wash or remaining bound to beads is shown by Western blotting, along with canonical Tip60 subunit Dmap1. (**E**) HA-tagged wild type, deacetylase domain 1 (HD1) mutant, or double deacetylase domain mutant (HD1 + HD2) Hdac6 was stably expressed in Tip60-H3F ESCs, Tip60-H3F was immunoprecipitated from nuclear extracts, and co-precipitating proteins were examined by Western blotting. Co-IP of canonical Tip60 complex subunit Dmap1 is shown as a control.

The following figure supplements are available for figure 5:

**Figure supplement 1**. *Hdac6* KD does not cause general disruption of Tip60-p400 complex.

**Figure supplement 2**. Tubastatin A has no effect on Hdac6 expression or localization.

**Figure supplement 3**. *Hdac6* KD reduces H2AK5 acetylation at Tip60-target genes.

*Figure 5. Continued on next page*

*Figure 5. Continued*

**Figure supplement 4**. *Hdac6* KD has no effect on bulk H4 acetylation levels, but strongly increases tubulin acetylation.

**Figure supplement 5**. *Hdac6* KD does not induce Tip60-p400 acetylation or ubiquitination.

model in which Hdac6 regulates gene expression by helping recruit Tip60-p400 complex to its targets in vivo. We tested this possibility by comparing Tip60 enrichment in control KD and *Hdac6* KD ESCs, predicting that if Hdac6 is necessary for Tip60-p400 recruitment, *Hdac6* KD should result in diminished Tip60 enrichment at many of its target genes. We observed a striking difference in Tip60 binding upon *Hdac6* KD compared to control cells: Tip60 enrichment downstream of the TSS was strongly reduced at many target genes in *Hdac6* KD ESCs, along with a modest decrease in enrichment upstream of the TSS (*Figure 6A–B*). Importantly, the reduction of Tip60 binding upon *Hdac6* KD was most severe at genes whose expression is regulated by Tip60-p400, compared to genes not regulated by the complex (*Figure 6C*, *Figure 6—figure supplement 1*). To confirm these findings, we tested the effects of *Hdac6* KD or mutation on the p400 subunit of Tip60-p400 complex, finding that *Hdac6* loss also strongly reduced p400 binding (*Figure 6D–E*, *Figure 6—figure supplement 2*).

Although we found that Hdac6 was required for Tip60 binding downstream of the TSS at most of its target genes, it remained possible that Tip60 was also required for Hdac6 to bind the same genes. To test this possibility, we mapped Hdac6 binding in control and *Tip60* KD ESCs. In contrast to Tip60's requirement for Hdac6, *Tip60* KD had no effect on Hdac6 localization (*Figure 6F*, *Figure 6—figure supplement 3*). Together, these data support a model in which Hdac6 binds to Tip60-p400 via its deacetylase domains, helping to recruit the complex to target promoters. In addition, these data rule out the possibility that Tip60-p400 represses differentiation genes by recruiting Hdac6 to these sites, since common targets of Tip60-p400 and Hdac6 were misregulated upon *Tip60* KD even though Hdac6 localization was unaffected.

## Hdac6 is required for normal ESC proliferation, colony formation, and robust differentiation but not self-renewal

Previously, we showed that ESCs depleted of Tip60-p400 subunits exhibited two prominent phenotypes: a failure to self-renew under conditions favoring ESC growth and a defect in the formation of embryoid bodies (EBs) under conditions favoring differentiation (*Fazzio et al., 2008a*). In contrast, while deletion of *Hdac6* in ESCs was reported to cause a defect in colony formation, as well as a slight proliferation defect, *Hdac6* KO ESCs could nonetheless continue to self-renew (*Zhang et al., 2003*), and mice derived from *Hdac6* KO ESCs are viable. However, it was unclear from previous reports whether *Hdac6* KO or KD might result in impaired ESC differentiation in vitro, similar to that of *Tip60* KD. To test this possibility, we first confirmed that *Hdac6* KD ESCs exhibited colony formation and proliferation defects similar to those described for *Hdac6* KO ESCs. Indeed, we observed a significant decrease in colony formation and size in both *Hdac6* KD and *Tip60* KD ESCs (*Figure 7A*). Furthermore, *Hdac6* KD ESCs had a small but reproducible decrease in proliferation rate, while *Tip60* KD caused a somewhat more severe proliferation defect (*Figure 7B*), consistent with previous studies (*Zhang et al., 2003*; *Fazzio et al., 2008a*). Next, we tested whether EB formation or cellular differentiation might be impaired upon *Hdac6* KD, as we previously observed upon KD of genes encoding known Tip60-p400 subunits *Tip60*, *p400*, or *Dmap1* (*Fazzio et al., 2008a*). EBs are thought to roughly mimic the embryonic state, as cells proliferate within spherical aggregates that subsequently develop cystic structures and differentiate into cells from all three germ layers (*Martin and Evans, 1975*). Indeed we found that, similar to *Tip60* KD ESCs, *Hdac6* KD ESCs formed EBs that were significantly smaller and more heterogeneous than control KD ESCs (*Figure 7C*). Furthermore, both *Hdac6* and *Tip60* KD EBs exhibited delayed induction of several differentiation markers during a time course of ESC differentiation (*Figure 8*). Thus, while *Hdac6* loss results in only modest phenotypes in self-renewing ESCs, it is necessary for normal EB formation and cellular differentiation in vitro, consistent with its role in Tip60-p400 recruitment to differentiation genes.

## Discussion

In this study, we showed that Hdac6 interacts with Tip60-p400 complex in ESCs, NSCs, and some cancer cell types, and is necessary for the proper regulation of most genes regulated by Tip60-p400

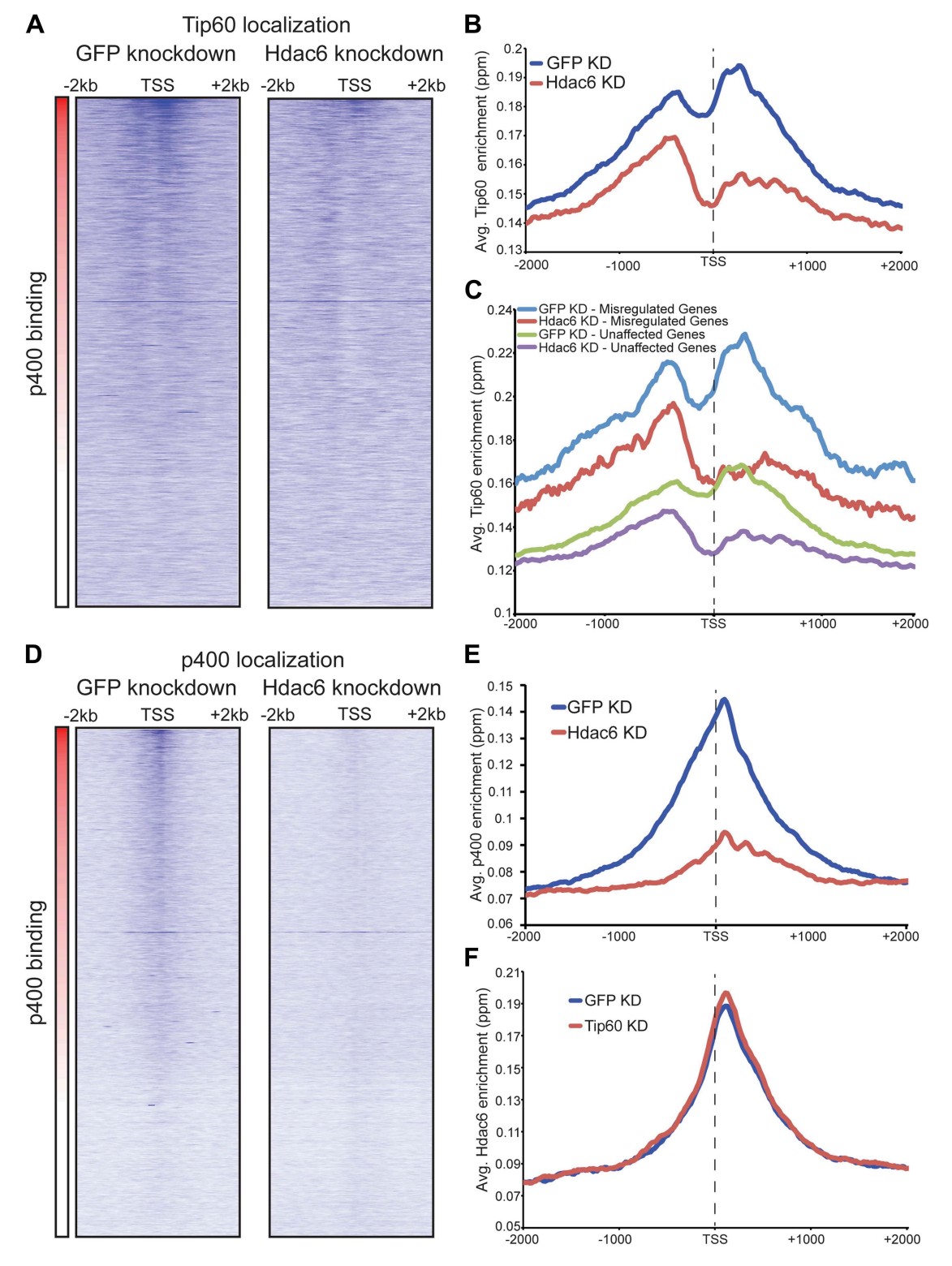

**Figure 6**. Hdac6 is necessary for normal Tip60 binding to its targets on chromatin. (**A**) Heat map representations of Tip60 binding as measured by ChIP-seq in Tip60-H3F ESCs upon *GFP* KD or *Hdac6* KD. Data are sorted by p400 binding as in *Figure 3A*. (**B**) Averaged Tip60 binding upon *GFP* KD or *Hdac6* KD are shown relative to the TSS. (**C**) *Hdac6* KD mainly reduces Tip60 enrichment at genes that are misregulated upon *Tip60* KD. Average Tip60 binding profiles upon *GFP* KD or *Hdac6* KD are shown for genes misregulated upon *Tip60* KD (adjusted p<0.1) and genes that are unaffected. (**D**) Heat
*Figure 6. Continued on next page*

*Figure 6. Continued*

map representations of p400 binding as measured by ChIP-seq using an antibody against endogenous p400 in control and *Hdac6* KD ESCs. Genes are sorted exactly as in (**A**). (**E**) Average p400 binding profiles upon *GFP* KD or *Hdac6* KD are shown relative to the TSS. (**F**) *Tip60* KD does not affect Hdac6 binding. Average Hdac6 binding profiles upon *GFP* KD or *Hdac6* KD are plotted as in (**B** and **E**).

The following figure supplements are available for figure 6:

**Figure supplement 1**. Reduced levels of Tip60 binding to differentiation genes upon *Hdac6* KD.

**Figure supplement 2**. Reduced levels of p400 binding to differentiation genes upon *Hdac6* KD.

**Figure supplement 3**. Hdac6 binding is unaffected by *Tip60* KD.

in ESCs. For a majority of Tip60-p400 target genes, we found that Hdac6 facilitated Tip60-p400 binding to its gene targets. Interestingly, while Hdac6 does not appear to deacetylate histones in vivo, we found that its catalytic domains were necessary for interaction with Tip60-p400 complex. Hdac6 has well established deacetylase activity directed against several cytoplasmic proteins, most notably tubulin (*Hubbert et al., 2002*), leaving open the possibility that, in stem cells, it may deacetylate some nuclear protein(s) that remain to be discovered.

Tip60-p400 binds chromatin near the 5′ ends of genes in two peaks surrounding the TSS: an upstream peak that does not overlap with Hdac6 and is moderately sensitive to its loss, and a downstream peak that overlaps with and more strongly requires Hdac6 (*Figure 9*). Recruitment of Tip60-p400 is the only apparent function of Hdac6 in gene regulation, since Hdac6 binding is maintained upon *Tip60* KD, while gene silencing is not. These data contrast with previously observed genetic interactions of mammalian Tip60 with other HDACs, in which Tip60 recruitment counteracts the repressive effects of HDACs on common targets (*Baek et al., 2002*), although similarities in gene expression profiles between Tip60 loss of function and chemical inhibition of HDACs have been reported in *Drosophila* (*Schirling et al., 2010*).

We found that *Hdac6* KD or mutation recapitulates most phenotypes of ESCs lacking Tip60 or p400: de-repression of many differentiation genes, impaired colony formation, a slower proliferation rate, impaired EB formation, and delayed kinetics of differentiation. Consistent with the phenotypes shared by *Hdac6* KD and *Tip60* KD ESCs, both *Hdac6* KD and loss-of-function mutations of Tip60-p400 subunits were previously shown to impair anchorage-independent growth and increase DNA damage sensitivity in cancer cells (*Feng et al., 2003*; *Lee et al., 2008*; *Wang et al., 2012*). However, unlike *Tip60* (*Hu et al., 2009*), *Hdac6* KO ESCs can self-renew and are competent for mouse development (*Zhang et al., 2008*), strongly suggesting that the defect observed for *Hdac6* KD ESCs in vitro is overcome in the embryo by compensatory mechanisms. Since Hdac6 is only partially required for Tip60-p400 function in ESCs, the levels to which Tip60 and Hdac6 target genes are misregulated within the ICM of *Hdac6*$^{-/-}$ embryos may not be severe enough to induce a developmental arrest. In addition, since Hdac6 appears to function within the nucleus in only a subset of cell types (stem and progenitor cells), the effect of *Hdac6* loss on gene regulation is likely much more limited than that of *Tip60* loss throughout the embryo.

The dynamic regulation of Hdac6 localization during stem cell differentiation suggests that nuclear exclusion of Hdac6 may play a major role in gene regulation by reducing Tip60-p400 binding to specific sets of genes in differentiated cells. Consistent with this model, Hdac6-dependent Tip60-target genes in ESCs were not bound by Tip60 in MEFs (*Figure 4—figure supplement 3*), suggesting that Hdac6 does not play any role in Tip60-p400 regulation in these cells. It remains to be determined whether alternative proteins substitute for Hdac6 in differentiated cells to recruit Tip60-p400 to different promoters, or if alternative mechanisms of recruitment (e.g., binding to histone modifications or interaction with sequence-specific DNA-binding proteins) are more important upon differentiation.

Together, these studies identify a new, stem cell-specific mechanism by which Tip60-p400 is regulated, and uncover a role for Hdac6 in gene regulation. These findings differ substantially from established models of Hdac6 function as mainly a regulator of cellular motility and clearance of misfolded proteins via deacetylation of a small set of cytoplasmic targets, and suggest that cell type must be carefully considered when examining the phenotypes observed upon *Hdac6* loss of function. In addition, these

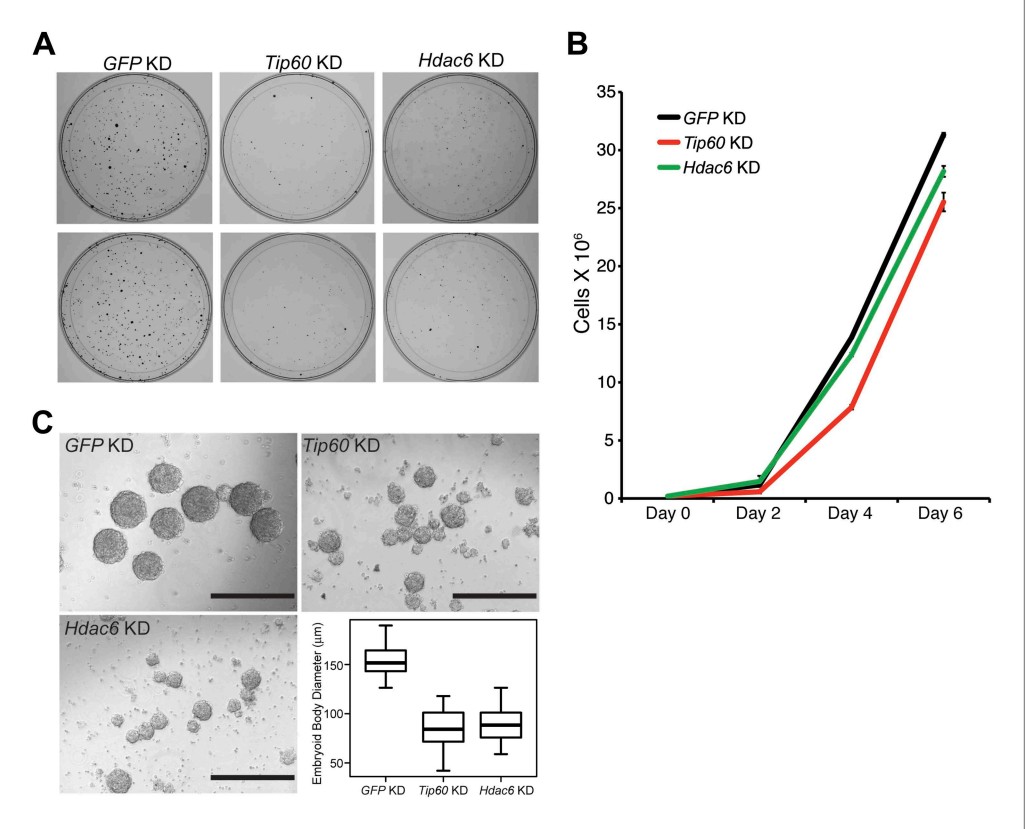

**Figure 7**. Hdac6 is necessary for ESC colony and EB formation. (**A**) Colony formation assays. Indicated KD ESCs were plated at clonal density and grown for seven days, at which time they were stained with crystal violet for visualization. (**B**) Growth curve. ESCs were infected with shRNA expressing viruses as shown and cultured in normal ESC medium. Cells were counted at the times indicated. (**C**) EB formation assay. Left and upper right: Brightfield images of EBs formed by hanging-drop cultures of ESCs knocked down as indicated, then cultured in differentiation medium, as described in 'Materials and methods'. Scale bars equal 400 μm. Lower right: box plot quantification of the range of EB sizes by diameter. The upper and lower limits of the box correspond to the 75th and 25th percentiles of each KD, respectively, and the dark line corresponds to the median of each box. At least 88 EBs were measured for each KD.

studies lend support to the idea that different types of stem cells, as well as some cancer cells, share common features of chromatin structure that allow them to maintain their developmental potency, and provide one possible mechanistic link underlying this commonality. Re-localization of Hdac6 to the cytoplasm during stem cell differentiation could lead to expression of an alternate set of Tip60-p400 target genes. Conversely, after Hdac6 is restored to the nucleus upon cellular transformation, it may re-direct Tip60-p400 to a subset of its target genes that are normally stem cell-specific, potentially eliciting changes in gene expression that promote cancer development.

## Materials and methods

### Cells

ESCs were grown under feeder-free conditions and used for all ESC experiments. The *Tip60-H3F* and *Hdac6-H3F* lines were made by targeting into E14 (*Hooper et al., 1987*) and *Hdac6^tm1a(EUCOMM)Wtsi* ESCs were obtained from EUCOMM (clone EPD0519_4_C03). This clone expresses low levels of Hdac6 protein before introduction of CRE, while CRE addition converts the mutation to a deletion. Therefore, we refer to the Hdac6 alleles as *Hdac6^reduced* and *Hdac6^null* before and after CRE introduction, respectively. CRE was introduced by Lenti-LucS (Addgene plasmid 22778) lentiviral infection, and the cells were harvested 3 days later for RT-qPCR or ChIP.

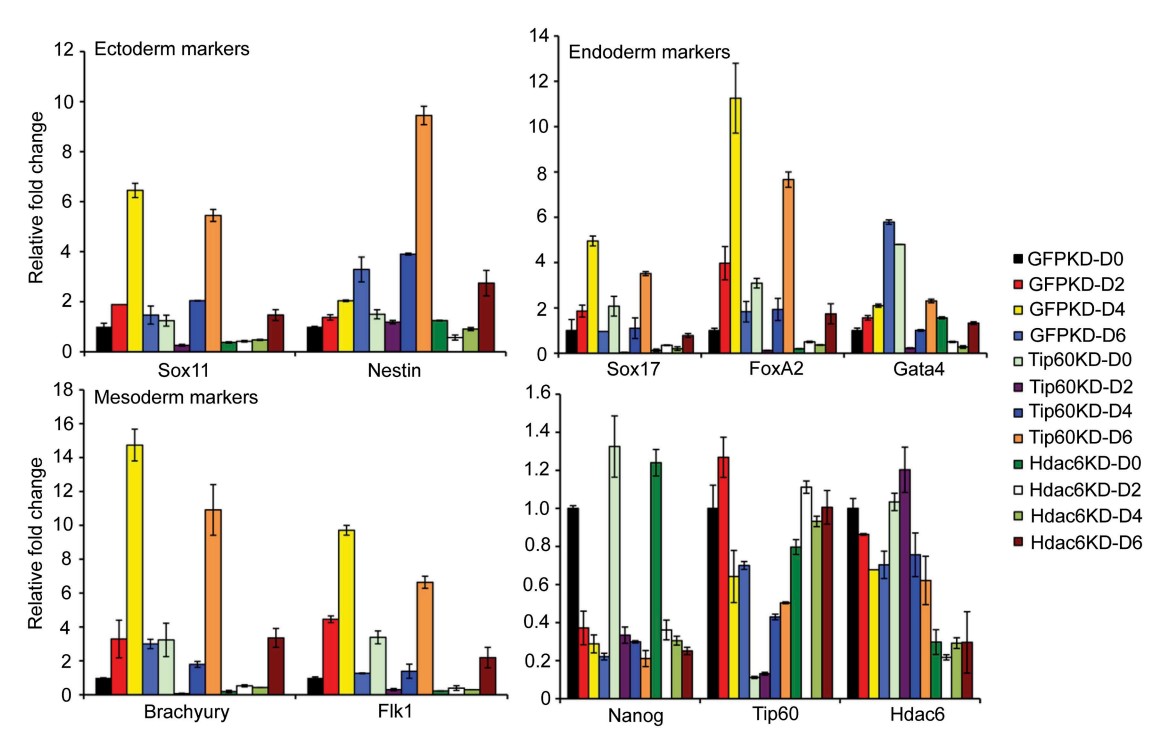

**Figure 8**. Impaired differentiation of *Hdac6* KD ESCs. Indicated KDs were differentiated for 0, 2, 4, or 6 days. At the indicated time points, RNA was isolated and RT-qPCR quantification of several differentiation markers of each primary germ layer was performed. Data shown are mean ± SD of three technical replicates from one representative experiment of two biological replicates performed.

Tip60-EGFP and Hdac6-HD mutant-HA expression: ES cells were infected with pLJM1puro lentiviral vectors containing mouse Tip60 cDNA fused to EGFP. The infected cells were selected by puromycin 3 days post-infection and Tip60 expression was checked by Western-blot. Hdac6 was cloned into pBabe HAII-hygro retroviral vector with or without previously described HD1 and HD2 mutations (*Zhang et al., 2006*). MEFs were infected with pBabe retroviral vectors containing wild-type or mutant mouse Hdac6 cDNAs. The infected cells were selected by hygromycin 3 days post-infection and Hdac6 expression was checked by Western-blot.

For imaging, embryoid bodies (EBs) were generated by placing 300 cells, after infection with shRNA-expressing lentiviruses as indicated, in 30 µl of medium lacking LIF and incubating in hanging-drop cultures. For examination of differentiation markers, $10^6$ cells were suspended in non-cell culture treated petri dishes for 2 days, and transferred to gelatin-coated cell culture dishes for another 4 days. RNA was isolated at the indicated time points. Mouse embryonic fibroblasts (MEFs) were isolated from *Tip60-H3F/+* or wild-type littermates using standard protocols (*Coles et al., 2007*) and mouse NSCs were isolated as previously described (*Li et al., 2008*).

Colony formation assay: control, *Tip60*, or *Hdac6* KD ESCs were plated 3 days after lentiviral shRNA infection. 2000 cells were seeded into each 10 cm dish and cultured for 7 days. The cells were fixed and stained with fixation-staining solution (6% glutaraldehyde, 0.5% crystal violet) for 30 min at room temperature followed by three washes with water.

For examination of Hdac6 redistribution during stem cell differentiation, ESCs were plated in N2B27 medium (*Ying et al., 2003*) and NSCs were plated in DMEM with 10% fetal bovine serum (FBS) for the number of days indicated. The cancer cell lines were grown in DMEM with 10% fetal bovine serum.

## RNAi

Lentiviral shRNA expression vectors from the TRC library (Thermo Fisher, Waltham, MA, USA) were obtained from the UMMS RNAi Core Facility. After screening through multiple shRNAs for each gene

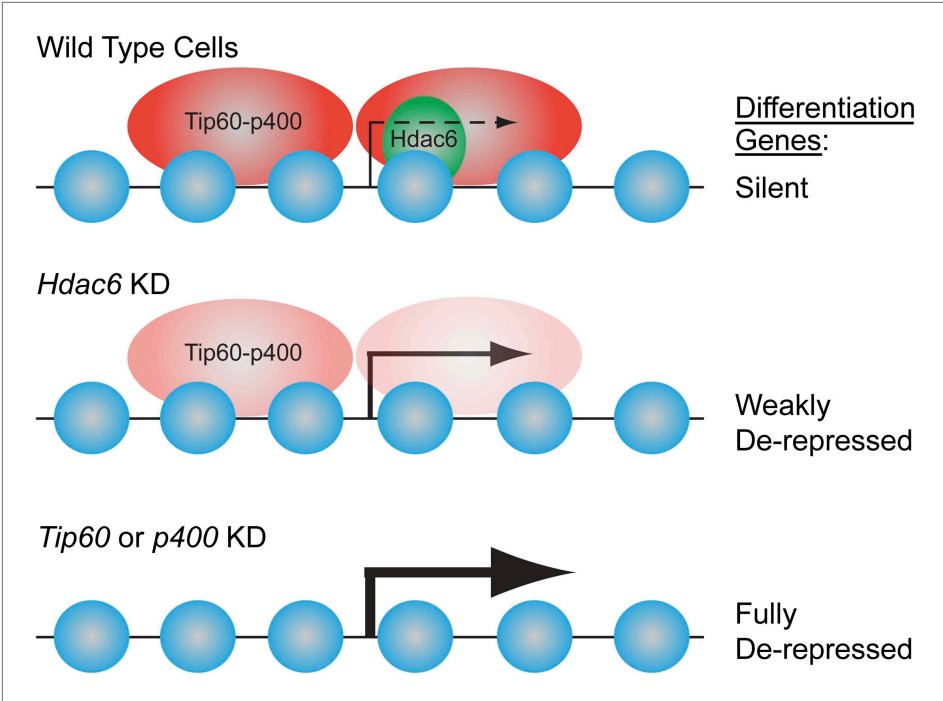

**Figure 9**. Model for Hdac6- and Tip60-p400-dependent repression of differentiation genes in ESCs. In the presence of Hdac6, Tip60-p400 binds both upstream and downstream of TSSs and differentiation genes are silenced. In the absence of Hdac6, Tip60-p400 binding is reduced causing de-repression of differentiation genes. Tip60-p400 binding downstream of target TSSs appears to be more strongly affected by *Hdac6* KD. Note that Hdac6 also appears to affect the activation of some genes (not depicted), also by recruitment of Tip60-p400 complex.

to be knocked down, we identified the most effective hairpin for each gene for subsequent experiments, listed as follows:

    pLKO.1/shGFP: GCAAGCTGACCCTGAAGTTCAT
    pLKO.1/shTip60: CGGAGTATGACTGCAAAGGTT
    pLKO.1/shHdac6: CGCTGACTACATTGCTGCTTT

Lentiviral vectors and Lenti-X packaging plasmids were transfected into 293.T cells using Fugene6 (Roche, Branford, CT, USA). At 48, 60 and 72 hr after transfection, lentivirus-containing media were collected and concentrated over a 20% sucrose cushion by centrifugation at 14,000 rpm for 4 hr in an SW-28 rotor. Concentrated virus was re-suspended in 200 µl PBS, aliquoted, and stored at −80°C.

## Western blotting and immunoprecipitation

Cells were lysed using an NE-PER Extraction kit (Thermo Fisher) to isolate cytoplasmic and nuclear fractions. Western blotting was performed with antibodies against Hdac6 (Cat. 07-732; Millipore, Billerica, MA, USA), acetyl-Histone H4 (Cat. 06-866; Millipore), β-actin (Cat. A5316; Sigma, St. Louis, MO, USA), Flag-M2 (Cat. F1804; Sigma), p400 (Cat. A300-541A; Bethyl Labs, Montgomery, TX, USA), Dmap1 (Cat. 10411-1-AP; Proteintech Group, Chicago, IL, USA), GFP (Cat. ab290; Abcam, Cambridge, MA, USA), Pol II (Cat. sc-899; Santa Cruz Biotechnology, Santa Cruz, CA, USA), Acetylated Lysine (Cat. 9441S; Cell Signaling Technologies, Danvers, MA, USA), ubiquitin (P4D1), and HA (12CA5). For immunoprecipitation, the aliquots of nuclear extract were incubated with specific antibodies conjugated with protein G magnetic beads (New England Biolabs, Ipswich, MA, USA) or FLAG-M2 Agarose beads (Sigma) in IP buffer (50 mM Tris-HCl pH7.4, 250 mM NaCl, 0.1% Triton X-100, plus 1X HALT protease inhibitors (Thermo Fisher) overnight at 4°C. To examine the effect of inhibitors of Hdac6 on its interaction with Tip60 complex, beads were subjected to five additional washes with or without 10 µM tubastatin A (ChemieTek, Indianapolis, IN, USA).

## Tip60-p400 purification

Tip60 complex was purified from nuclear extracts of Tip60-H3F ESCs as described previously for Mbd3 (*Yildirim et al., 2011*). Briefly, nuclear extracts were subjected to sequential affinity purification steps using FLAG-M2 Agarose (Sigma) and TALON Agarose beads (Clontech Laboratories, Mountain View, CA, USA). The proteins purified from untagged control and Tip60-H3F cells were separated by SDS-PAGE and were either stained with SimplyBlue SafeStain (Invitrogen, Grand Island, NY, USA) after TCA precipitation and re-suspension in sample buffer or SilverXpress (Invitrogen) for visualization in *Figure 1*.

## LC-MS/MS

Affinity-purified samples were separated by SDS-PAGE gel, in-gel digested and analyzed by LC-MS and LC-MS/MS as described previously (*Chu et al., 2006*). Briefly, 1 ml aliquot of the digestion mixture was injected into a Dionex Ultimate 3000 RSLCnano UHPLC system with an autosampler (Dionex Corporation, Sunnyvale, CA, USA), and the eluant was connected directly to a nanoelectrospray ionization source of an LTQ Orbitrap XL mass spectrometer (Thermo Fisher). LC-MS data were acquired in an information-dependent acquisition mode, cycling between a MS scan (m/z 310-2000) acquired in the Orbitrap, followed by low-energy CID analysis in the linear ion trap. The centroided peak lists of the CID spectra were generated by PAVA (*Guan and Burlingame, 2010*) and searched against a database that consisted of the Swiss-Prot protein database, to which a randomized version had been concatenated, using Batch-Tag, a program in the in-house version of the University of California San Francisco Protein Prospector version 5.9.2. A precursor mass tolerance of 15 ppm and a fragment mass tolerance of 0.5 Da were used for protein database search. Protein hits are reported with a Protein Prospector protein score ≥22, protein discriminant score ≥0.0 and a peptide expectation value ≤0.01 (*Chalkley et al., 2005*). This set of protein identification parameters threshold did not return any substantial false positive protein hit from the randomized half of the concatenated database.

## Chromatin immunoprecipitation

The cells from ~80% confluent 10 cm dishes were crosslinked by adding fixation solution (1% formaldehyde, 0.1M NaCl, 1 mM EDTA, 50 mM HEPES·KOH pH 7.6) for 10 min at room temperature. Crosslinking was quenched with 125 mM Glycine for 5 min. The cells were washed twice with cold PBS containing protease inhibitors (Roche), and pelleted at 1000×*g* for 5 min at 4°C. The cell pellets were either flash frozen in liquid nitrogen and stored at −80°C or immediately sonicated. The pellets were resuspended in Lysis buffer 1 (50 mM HEPES·KOH pH 7.6, 140 mM NaCl, 1 mM EDTA, 10% (vol/vol) Glycerol, 0.5% NP-40, 0.25% Triton X-100) including protease inhibitors and incubated for 10 min at 4°C. After centrifugation at 1350×*g* for 5 min, the pellets were resuspended in Lysis buffer 2 (10 mM Tris-HCl pH 8.0, 200 mM NaCl, 1 mM EDTA, 0.5 mM EGTA) containing protease inhibitors and incubated for another 10 min at 4°C. The pellets were collected after centrifugation at 1350×*g* for 5 min and resuspended in Lysis buffer 2 for sonication. The samples were sonicated in a Bioruptor (UCD-200; Diagenode, Delville, NJ, USA) set to high for three cycles (10 min per cycle with 30 s on/ 30 s off) to generate 300–1000 base-pair fragments. The supernatants were collected after a 13,000 rpm spin for 10 min at 4°C. 50-μl Protein G Magnetic beads (New England Biolabs) were washed twice with PBS with 5 mg/ml BSA and 10 μg of anti-Flag M2 antibody (Sigma) coupled in 500 μl PBS with 5 mg/ml BSA overnight at 4°C. Immunoprecipitation was performed with antibody-coupled beads and sonicated supernatants in ChIP buffer (20 mM Tris-HCl pH8.0, 150 mM NaCl, 2 mM EDTA, 1% Triton X-100) overnight at 4°C. The magnetic beads were washed twice with ChIP buffer, once with ChIP buffer including 500 mM NaCl, four times with RIPA buffer (10 mM Tris-HCl pH8.0, 0.25M LiCl, 1 mM EDTA, 0.5% NP-40, 0.5% Na·Deoxycholate), and once with TE buffer (pH 8.0). Chromatin was eluted twice from washed beads by adding elution buffer (20 mM Tris-HCl pH8.0, 100 mM NaCl, 20 mM EDTA, 1% SDS) and incubating for 15 min at 65°C. The crosslinking was reversed at 65°C for 6 hr and RNase A (Sigma) was added for 1 hr at 37°C followed by proteinase K (Ambion, Carlsbad, CA, USA) treatment overnight at 50°C. ChIP-enriched DNA was purified using Phenol/Chloroform/Isoamyl alcohol extractions in phase-lock tubes. Then, chromatin was analyzed by qPCR using a SYBR FAST universal kit (KAPA Biosystems, Woburn, MA, USA) with specific primers (*Supplementary file 1A*).

## ChIP-seq

### Library construction

Chromatin immunoprecipitation and deep sequencing library construction were performed using minor modifications of our chromatin immunoprecipitation protocol. The samples were crosslinked

and prepared as described previously (*Yildirim et al., 2011*). The samples were transferred into 15 ml Falcon tubes and sonicated in a Bioruptor set to high for two cycles (10 min for one cycle with 30 s on/30 s off). ChIP samples were end-repaired, A-tailed, and adaptor-ligated using barcoded adaptors according to the manufacturer's instructions (Illumina, San Diego, CA, USA). DNA purification on Zymo Research PCR purification columns was performed following each enzyme reaction (Zymo Research, Irvine, CA, USA). The adaptor-ligated material was then PCR amplified with Phusion polymerase using 16 cycles of PCR before size selection of 200–300 bp fragments on a 2% agarose gel. The library was purified using a Zymo Gel Extraction kit, its concentration was determined using a NanoDrop (Thermo), and the integrity of each library was confirmed by sequencing 10–20 individual fragments per library. Libraries with different barcodes were pooled together and single-end sequencing (50 bp) was performed on an Illumina HiSeq2000 at the UMass Medical School deep sequencing core facility.

### Data analysis

Raw FastQ reads were first collapsed by sequence and the read occurrences were kept. The reads were then mapped to the mm9 genome using bowtie allowing at most one mismatch in every alignment. For multimappers, only one alignment was chosen randomly by the M 1 parameter setting. Each aligned location was extended downstream to a length of 150 bp. Any extension that exceeded the end of the chromosome was clipped. The extended mapped locations overlapping with simple repeats annotated by RepBase were removed. For each remaining read along with its occurrence, we calculated the relative distance to the nearest TSS and for each TSS tallied the sum of read occurrence from its upstream 2000 bp to downstream 2000 bp. The occurrences were normalized and binned in 20 bp intervals. Deep sequencing data can be obtained from GEO (http://www.ncbi.nlm.nih.gov/geo/), accession: GSE42329.

### Microarray analysis

5 µg of total RNA from control (GFP), *Tip60*, *Hdac6* KD or double KD ESCs was subjected to RNA amplification and labeling using the Low Input Quick Amp Labeling Kit protocol (Agilent, Santa Clara, CA, USA) with minor modifications. Briefly, cRNA was amplified by in vitro transcription with amino-allyl UTP (3:2 ratio for amino-ally UTP: UTP) overnight at 37°C. Then, cRNA was purified using Zymo RNA purification columns and labeled with Cy3 (GE Healthcare, Uppsala, Sweden) at room temperature for 60 min in the dark. The fluorescence intensity of Cy3 was determined by NanoDrop and 50 picomoles of cRNA was used for fragmentation and hybridization on Agilent 4X44K mouse whole-genome microarrays. Slides were scanned on Agilent DNA microarray scanner G2565CA and fluorescence data were obtained using Agilent Feature Extraction software at the UMass Medical School genomics core facility. The expression profiles from two biological replicates were analyzed as previously described (*Yildirim et al., 2011*). Enrichment of Gene Ontology terms and categories was performed with DAVID 6.7 (*Huang et al., 2009a, 2009b*). Microarray data can be obtained from GEO (http://www.ncbi.nlm.nih.gov/geo/), accession: GSE42329.

## Acknowledgements

We thank J Benanti, O Rando, S Hainer, and L Ee for critical reading of the manuscript. We also thank J Benanti, M Green, D Guertin, N Kalaany, and P Matthias for reagents and cell lines; and J Rodriguez and T Tsukiyama for technical advice.

## Additional information

### Funding

| Funder | Grant reference number | Author |
| --- | --- | --- |
| National Institutes of Health | HD072122 | Thomas G Fazzio |
| Pew Charitable Trusts | | Thomas G Fazzio |
| National Institute of Food and Agriculture/US Department of Agriculture | H00567 | Feixia Chu |
| National Science Foundation | DBI-0850008 | Zhiping Weng |

The funders had no role in study design, data collection and interpretation, or the decision to submit the work for publication.

## Author contributions

PBC, TGF, Conception and design, Acquisition of data, Analysis and interpretation of data, Drafting or revising the article; J-HH, Analysis and interpretation of data, Drafting or revising the article; TLH, AHC, Acquisition of data, Analysis and interpretation of data; JFC, Acquisition of data, Contributed unpublished essential data or reagents; ZW, Conception and design, Analysis and interpretation of data; FC, Conception and design, Acquisition of data, Analysis and interpretation of data

## Ethics

Animal experimentation: This study was performed in strict accordance with the recommendations of the Institutional Animal Care and Use Committee at the University of Massachusetts Medical School (approval #2165-13).

## Additional files

### Supplementary files

• Supplementary file 1. A. ChIP-qPCR primers used. B. RT-qPCR primers used.

### Major dataset

The following dataset was generated:

| Author(s) | Year | Dataset title | Dataset ID and/or URL | Database, license, and accessibility information |
| --- | --- | --- | --- | --- |
| Chen P | 2013 | Hdac6 is a stem-cell specific modulator of Tip60-p400 function | GSE42329; http://www.ncbi.nlm.nih.gov/geo/query/acc.cgi?token=tvmhfusekaaakpy&acc=GSE42329 | Publicly available at NCBI GEO. |

The following previously published datasets were used:

| Author(s) | Year | Dataset title | Dataset ID and/or URL | Database, license, and accessibility information |
| --- | --- | --- | --- | --- |
| Mikkelsen TS, Ku M, Lander ES, Bernstein BE | 2008 | Genome-wide maps of chromatin state in pluripotent and lineage-committed cells | GSE12241; http://www.ncbi.nlm.nih.gov/geo/query/acc.cgi?token=tvmhfusekaaakpy&acc=GSE12241 | Publicly available at NCBI GEO. |
| Hu G, Cui K, Northrup D, Liu C, Wang C, Tang Q, Ge K, Levens D, Crane-Robinson C, Zhao K | 2012 | H2A.Z Facilitates Access of Active and Repressive Complexes to Chromatin in Embryonic Stem Cell Self-renewal and Differentiation | GSE34483; http://www.ncbi.nlm.nih.gov/geo/query/acc.cgi?token=tvmhfusekaaakpy&acc=GSE34483 | Publicly available at NCBI GEO. |
| Teif VB, Vainshtein Y, Caudron-Herger M, Mallm J, Marth C, Hofer T, Rippe K | 2012 | Genome-wide nucleosome positioning during embryonic stem cell development | GSE40896; http://www.ncbi.nlm.nih.gov/geo/query/acc.cgi?token=tvmhfusekaaakpy&acc=GSE40896 | Publicly available at NCBI GEO. |

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
