## [Decision Letter]

Thank you for sending your work entitled “Hdac6 regulates Tip60-p400 function in stem cells” for consideration at *eLife*. Your article has been favorably evaluated by a Senior editor and 2 reviewers, one of whom is a member of our Board of Reviewing Editors, and one of whom, Jerry Workman, has agreed to reveal his identity. The Reviewing editor has assembled the following comments to help you prepare a revised submission.

This is a very nice paper that deserves to be published in *eLife*. The paper shows that Hdac6 behaves differently in stem cells (and some de-differentiated cancer cells) by being nuclear and interacting with the Tip60-p400 complex. This interaction with the Tip60-p400 complex, which is surprisingly mediated by the histone deacetylase domain, is necessary for Hdac6 to recruit the Tip60-p400 complex to selected promoters and inhibit transcription. Biologically, this interaction is important for stem cell differentiation. Overall, this paper reports surprising findings, breaks new ground both with respect to histone acetylase and deacetylase complexes in a context related to stem cells. The experiments are well done, the results and conclusions convincing, and the paper well written. The manuscript is essentially accepted, but please look at the comments below and make any modifications that you wish. The paper does not need to be re-reviewed.

1) It would be interesting to examine the acetylation status of Tip60-p400 more thoroughly. For example, they might want to immunoblot their purifications done in Hdac6 WT vs mutant backgrounds with an anti-acetyl-lysine antibody. Or, they could do metabolic labeling with 3H acetate if the antibody is not adequate.

2) Hdac6 also binds ubiquitin. Does treatment with the drug affect Hdac6 binding to ubiquitin? More importantly, are any of the complex members ubiquitinated?

3) It is not clear how Hdac6 might recruit Tip60-p400 to chromatin. *Hdac6* has been shown to regulate ubiquitination of proteins through its ability to bind and protect elongating polyubiquitin chains. The authors should determine whether loss of *Hdac6* alters Tip60-p400 complex ubiquitination and/or degradation. They could also determine whether mg132 treatment can partially rescue Tip60-p400 chromatin localization in the absence of Hdac6. Figure 5—figure supplement 1 appears to show that Tip60 shifts up in size.

4) It might be possible that Hdac6 binds to chromatin though ubiquitinated histones and then recruits Tip60-p400. The data show that Hdac6 is associated with H3K4me2 and H3K4me3 and these marks have been associated with H2B ubiquitination, although the pattern of Hdac6 localization looks more like the pattern of H2A ubiquitination implied by Endoh et al. Does knockdown of a histone ubiquitin ligase reduce Hdac6 chromatin localization? Or does mutation of the Hdac6 ubiquitin-binding domain do the same?

---

## [Author Response]

*1) It would be interesting to examine the acetylation status of Tip60-p400 more thoroughly. For example, they might want to immunoblot their purifications done in Hdac6 WT vs mutant backgrounds with an anti-acetyl-lysine antibody. Or, they could do metabolic labeling with 3H acetate if the antibody is not adequate*.

We now include anti-acetyl-lysine blots of complex purified from control or *Hdac6* KD ESCs in Figure 5—figure supplement 5. We did not observe detectable acetylation of Tip60-p400 complex in the presence or absence of *Hdac6* KD. Therefore, if Hdac6 deacetylates any subunits of the complex, their acetylation levels must be below the level of detection of the anti-acetyl-lysine antibody, even when *Hdac6* is knocked down.

*2) Hdac6 also binds ubiquitin. Does treatment with the drug affect Hdac6 binding to ubiquitin? More importantly, are any of the complex members ubiquitinated*?

To address whether subunits of Tip60-p400 complex are ubiquitinated, we Western blotted complex purified from control or *Hdac6* KD ESCs with an anti-ubiquitin antibody, and found no detectable ubiquitination in purified complex (Figure 5—figure supplement 5). These data indicate that if subunits of Tip60-p400 complex are ubiquitinated, the levels of ubiquitination in the cell are below the limit of detection of the anti-ubiquitin antibody, suggesting that this is not the mechanism by which Hdac6 binds Tip60-p400.

*3) It is not clear how Hdac6 might recruit Tip60-p400 to chromatin.* Hdac6 *has been shown to regulate ubiquitination of proteins through its ability to bind and protect elongating polyubiquitin chains. The authors should determine whether loss of* Hdac6 *alters Tip60-p400 complex ubiquitination and/or degradation. They could also determine whether mg132 treatment can partially rescue Tip60-p400 chromatin localization in the absence of Hdac6.*
Figure 5—figure supplement 1
*appears to show that Tip60 shifts up in size*.

To address whether loss of Hdac6 alters stability or levels of the complex, we silver stained Tip60-p400 complex purified from control or *Hdac6* KD ESCs and added this gel to Figure 5—figure supplement 1. We did not observe a reduction in the majority of subunits in the complex, suggesting that the levels and stability of most subunits are maintained in the absence of *Hdac6* (consistent with Western blotting data in Figure 3 and Figure 5—figure supplement 1). However, we did find that one prominent band (besides Hdac6) was reduced upon *Hdac6* KD (Figure 5—figure supplement 1). These data suggest that Hdac6 is not necessary for levels of Tip60-p400 complex, but is required for interaction of at least one protein with the complex. It is not clear if this protein is subject to ubiquitin-mediated degradation in the absence of Hdac6. Additional studies will be necessary to determine the identity of this protein so that we can explore this possibility further.

Regarding Tip60 mobility, we don’t believe we see a mobility difference upon *Hdac6* KD. In Figure 5—figure supplement 1, Tip60-H3F levels are slightly higher in the *Hdac6* KD/Tip60-H3F IP lane. Multiple isoforms of Tip60 are expressed in ESCs, all of which harbor the H3F tag at the C-terminus. The two lowest mobility bands in Figure 5—figure supplement 1 blur together a bit in the Hdac6 KD lane, perhaps giving the illusion of a mobility shift. This pattern is observed in additional blots of Tip60-H3F in the manuscript (Figures 3 and 5), which show no difference in Tip60 mobility or levels upon *Hdac6* KD or inhibition, respectively.

*4) It might be possible that Hdac6 binds to chromatin though ubiquitinated histones and then recruits Tip60-p400. The data show that Hdac6 is associated with H3K4me2 and H3K4me3 and these marks have been associated with H2B ubiquitination, although the pattern of Hdac6 localization looks more like the pattern of H2A ubiquitination implied by Endoh et al. Does knockdown of a histone ubiquitin ligase reduce Hdac6 chromatin localization? Or does mutation of the Hdac6 ubiquitin-binding domain do the same*?

We agree that a thorough examination of which domains of Hdac6 are necessary for chromatin association and which chromatin modifications or transcription factors recruit Hdac6 to chromatin will be of significant interest. However, these experiments will require considerable time and effort to perform in a comprehensive manner. Because of this, and the fact that we were primarily interested in understanding how Hdac6 interacts with and regulates Tip60-p400 in this manuscript, we believe these additional studies would be better suited as the focus of a separate study.